# New particle formation events observed at the King Sejong Station, Antarctic Peninsula – Part 2: Link with the oceanic biological activities

Eunho Jang[1, 2,*], Ki-Tae Park[1,*], Young Jun Yoon[1], Tae-Wook Kim[3], Sang-Bum Hong[1], Silvia Becagli[4], Rita Traversi[4], Jaeseok Kim[5], Yeontae Gim[1]

[1]Korea Polar Research Institute, 26 Songdomirae-ro, Yeonsu-gu, Incheon 21990, South Korea
[2]University of Science and Technology, 217 Gajeong-ro, Yuseong-gu, Daejeon 34113, South Korea
[3]Division of Environmental Science and Ecological Engineering, Korea University, Seoul, South Korea
[4]Department of Chemistry "Ugo Schiff", University of Florence, via della Lastruccia, 3, Sesto F.no (FI), 50019, Italy.
[5]Korea Research Institute of Standards and Science, 267 Gajeong-ro, Yuseong-gu, Daejeon 34113, South Korea
[*]These authors contributed equally to this work

*Correspondence to*: Ki-Tae Park (ktpark@kopri.re.kr)

**Abstract.** Marine biota is an important source of atmospheric aerosol particles in the remote marine atmosphere. However, the relationship between new particle formation and marine biota is poorly quantified. Long-term observations (from 2009 to 2016) of the physical properties of atmospheric aerosol particles measured at the Antarctic Peninsula (King Sejong Station; 62.2°S, 58.8°W) and satellite-derived estimates of the biological characteristics were analyzed to identify the link between new particle formation and marine biota. New particle formation events in the Antarctic atmosphere showed distinct seasonal variations, with the highest values occurred when the air mass originated from the ocean domain during productive austral summer (December, January and February). Interestingly, new particle formation events were more frequent in the air masses that originated from the Bellingshausen Sea than in those that originated from the Weddell Sea. The monthly mean number concentration of nanoparticles (2.5–10 nm in diameter) was >2-fold when the air masses passed over the Bellingshausen Sea than the Weddell Sea, whereas the biomass of phytoplankton in the Weddell Sea was more than ~70% higher than that of the Bellingshausen Sea during the austral summer period. Dimethyl sulfide (DMS) is of marine origin and its oxidative products are known to be one of the major components in the formation of new particles. Both satellite-derived estimates of the biological characteristics (dimethylsulfoniopropionate (DMSP; precursor of DMS) and phytoplankton taxonomic composition) and in situ methanesulfonic acid (84 daily measurements during the summer period in 2013 and 2014) analysis revealed that DMS(P)-rich phytoplankton were more dominant in the Bellingshausen Sea than in the Weddell Sea. Furthermore, the number concentration of nanoparticles was positively correlated with the biomass of phytoplankton during the period when DMS(P)-rich phytoplankton predominate. These results indicate that oceanic DMS emissions could play a key role in the formation of new particles; moreover, the taxonomic composition of phytoplankton could affect the formation of new particles in the Antarctic Ocean.

**1 Introduction**

Aerosols in the atmosphere significantly influence radiative forcing directly (by scattering incoming radiation) and indirectly (by modifying cloud micro-physical properties) (IPCC, 2013). Considering that the ocean surface accounts for approximately 70% of the total surface of the Earth, the marine aerosols are globally one of the most

important natural aerosol systems (O'Dowd et al., 2004; Leck and Bigg, 2005; O'Dowd and De Leeuw, 2007). Marine aerosols consist of primary and secondary aerosols. The primary aerosols are produced via the bubble bursting process and include mostly sea-salts and organic matter (O'Dowd and De Leeuw, 2007). Secondary aerosols (resulting from gas-to-particle conversion) are produced in the marine atmosphere through several different processes (e.g., binary, ternary and ion-induced) (Kulmala et al., 1990; Korhonen et al., 1999; Lee et al.,

2003; Lovejoy et al., 2004; Kirkby et al., 2016; Jokinen et al., 2018). Biogenic volatile organic compounds emitted into the marine atmosphere through air-sea gas exchange, and their oxidative products can be involved in new particle formation events via the nucleation of stable clusters of the order of 0.5–1 nm in size (O'Dowd and De Leeuw, 2007).

In the Antarctic, most submicron aerosol particles are derived from natural sources rather than from long-range

transport of anthropogenic sources (Ito, 1989; Kyrö et al., 2013). The Southern Ocean has been considered as the most significant source of secondary organic aerosols in the Antarctic atmosphere, especially during the phytoplankton bloom period (Asmi et al., 2010; Yu and Luo, 2010). Recent studies reported that biological (mostly derived by sea-ice algae) and abiotic (photochemical reaction of halogen compounds) processes occurring near sea-ice regions could significantly influence the formation and growth of aerosol particles (Atkinson et al.,

2012; Kyrö et al., 2013; Allan et al., 2015; Dall'Osto et al., 2017 and 2018). Molecular level evidence of new particle formation via sequential addition of iodine-containing species, which could be emitted from both open water and sea-ice zone, was reported in the Arctic and Antarctic sites (Sipilä et al., 2016). In addition, the emission of ammonia from seabird colonies in polar regions could act as a key factor contributing to bursts of newly formed particles (Weber et al., 1998; Croft et al., 2016). The Antarctic Peninsula is one of the three areas of the globe

facing drastic regional warming, and it undergoes rapid environmental changes (i.e., increased temperature, acidification, shallowing mixed layer depth, sea-ice decline, increased light, increased nutrient supply, reduced salinity and glacial retreat) (Clarke et al., 2007; Deppeler and Davidson, 2017). Such environmental changes in this region could significantly affect the marine ecosystem. Marine phytoplankton influence aerosol properties by releasing various organic compounds back into the atmosphere. The Antarctic Ocean is known to be a significant

source of numerous biogenic volatile organic compounds including dimethyl sulfide (DMS), organic nitrogen and halogenated organic compounds (Laturnus et al., 1996; Beyersdorf et al., 2010; Dall'Osto et al., 2017; Giordano et al., 2017). In particular, the Antarctic Ocean is the region with the highest sea-surface DMS concentration (Lana et al., 2011). Observations of both DMS and its oxidative products (i.e., methanesulfonic acid (MSA) and non-sea-salt $SO_4^{2-}$) in Antarctica show a clear seasonal cycle with a minimum in austral winter and a maximum in

austral summer (Prospero et al., 1991; Minikin et al., 1998; Preunkert et al., 2007; Read et al., 2008). DMS is produced by a complex marine biota food-web mechanism (Stefels et al., 2007; Park et al., 2014a). Phytoplankton produce dimethylsulfoniopropionate (DMSP) and then some of the DMSP is converted into gaseous DMS through enzymatic cleavage (Simó, 2001). The biogenic emission of DMS from the ocean is the largest natural sulfur source to the atmosphere (Andreae, 1990) and, the oxidation of DMS in the marine atmosphere is a key process

contributing to the formation of new particles (Park et al., 2017). Once the DMS in the sea-surface is emitted into the atmosphere, it is rapidly converted into $SO_2$ and MSA through the photochemical oxidation process. The $SO_2$ and MSA formed from DMS tend to contribute to the formation of new particles via nucleation processes and eventually serve as nuclei for cloud formation (Charlson et al., 1987). Secondary organic aerosols contribute to a large fraction of the submicron aerosol mass in the atmosphere, affecting clouds and climate (Jimenez et al., 2009). However, secondary organic aerosols are a source of considerable uncertainty in understanding current climate change (Shiraiwa et al., 2017). In particular, the formation of new particles in the remote marine atmosphere and their association with marine biota remains poorly quantified.

In this study, we aimed to identify the link between new particle formation and marine biota at a remote Antarctic site, where biological productivity is the highest in the global ocean. To this end, we analyzed the physical and chemical properties of aerosol particles at King Sejong Station (62.2°S, 58.8°W) on the Antarctic Peninsula from 2009 to 2016. To study the oceanic biological characteristics surrounding the observation site, we analyzed satellite-derived estimates including the chlorophyll concentration, total DMSP concentration and taxonomic composition of marine phytoplankton.

## 2 Experimental Methods

### 2.1. Aerosol measurements

King Sejong Station (62.2°S, 58.8°W) is located on the Antarctic Peninsula where severe climate change is occurring. The aerosol observatory is approximately 400 m southwest of the main facilities of King Sejong Station and approximately 10 m above sea level (m.a.s.l). Continuous observations of the physical properties of aerosol particles in the Antarctic atmosphere were conducted between March 2009 and December 2016 at the observatory. The number concentration of aerosol particles (CN) was measured using two condensation particle counters (CPCs) that have different measurement range limits for particle diameter: particles larger than 2.5 nm ($CN_{2.5}$; TSI model 3776) and particles larger than 10 nm ($CN_{10}$; TSI model 3772) (Fig. 1a). The difference in the particle number concentration between the two CPCs (i.e., particles between 2.5 and 10 nm in diameter) was interpreted as an indication of the existence of newly formed nanoparticles ($CN_{2.5-10}$) (Fig. 1b). An Aethalometer (Magee Scientific model AE16) was used to analyze the concentration of black carbon by measuring light-absorbing particles at the 880 nm wavelength. To avoid local influence on the aerosol properties, data with a black carbon concentration of >100 ng were excluded. Data were also excluded when the wind direction was between 355° and 55°. This is because the north-eastern direction is designated the local air pollution sector due to emissions from the power generators and crematory. More detailed information regarding the aerosol particle analysis at King Sejong Station is provided by Kim et al. (2017 and 2018).

For analysis of chemical properties of aerosol particle, an air sampler equipped with a $PM_{10}$ impactor (collecting particles <10 µm in aerodynamic equivalent diameter) was used to collect aerosol particles. The sampler was mounted on the roof of the aerosol observatory and sampled particles every 24 hours during the summer period in 2013 and 2014 (explicitly, from 14 January to 28 February in 2013, and from 2 December 2013 to 18 January 2014). Half of a 47-mm Teflon filter was used to measure major ions including MSA. The MSA (and the other ions) collected on the filter was extracted into about 10 mL (18 MΩ Milli-Q) in ultrasonic bath for 20 min. MSA

was determined by an ion chromatography system (Dionex, Thermo Fisher Scientific Inc.) following the procedure described by Becagli et al. (2012). For MSA, reproducibility on real samples was better than 5%. Filter blank concentrations for methanesulphonate were always below the detection limit.

## 2.2. Air-mass back trajectories

The air mass back trajectories and meteorological parameters were obtained using the Hybrid Single-Particle Lagrangian Integrated Trajectories model (Draxler and Hess, 1998). In general, the growth rate of sub-micron particles in the remote marine environment ranges from 0.2 to 5.0 nm h$^{-1}$ (Järvinen et al., 2013; Weller et al., 2015; Kerminen et al., 2018), and the mean growth rate of the aerosol particles measured at the King Sejong Station was approximately $0.7 \pm 0.3$ nm h$^{-1}$ during the eight years (Kim et al., 2018). Therefore, the 2-day air mass back trajectories and hourly positions were determined and combined with satellite-derived geographical information to identify the travel history of the air mass arriving at the observation site. Daily geographical information on sea-ice, land and ocean area was obtained from the Sea Ice Index at a 25-km resolution provided by the National Snow and Ice Data Center (NSIDC). The oceanic region adjacent to the observation site was surrounded by two different ocean basins, namely, the Bellingshausen and Weddell seas. To evaluate the influence of the oceanic biological characteristics on the occurrence of new particle formation, we limited our analysis to the air masses that had exposure predominantly to the ocean area. Specifically, the origin of the hourly air mass arriving at the observation site was divided into two ocean domains (i.e., the Bellingshausen and Weddell seas). Then, all air mass back trajectories were grouped into one of the two ocean domains by only selecting the 2-day air mass back trajectories that had >90% retention in a given ocean domain.

A total of 84 PM$_{10}$ samples for MSA analysis were collected daily during the summer periods in 2013 and 2014. The retention time of the aerosol particles with a diameter <10 µm is known to be approximately 3–5 days in the atmosphere (Mishra et al., 2004; Budhavant et al., 2015). Therefore, 3-, 4-, and 5-day air mass back trajectories were applied to identify the potential origin of MSA during the sampling period.

## 2.3. Phytoplankton biomass, DMSP and taxonomic composition analysis

Satellite-derived ocean color provides a good measure of analyzing the phytoplankton characteristics of the Southern Ocean (Siegel et al., 2013; Haentjens et al., 2017). The phytoplankton biomass of the two ocean domains was estimated by calculating the chlorophyll concentration from the Moderate Resolution Imaging Spectroradiometer on the Aqua (MODIS-Aqua) satellites at 4 km resolution during the study period (2009–2016). The trajectory concentration of the air masses originating from the two ocean domains was calculated from the ratio of the number of hourly trajectory points passing over each grid cell ($1° \times 1°$) to the total number of hourly trajectory points (Kim et al., 2011), as shown in Fig. 2a. We limited our analysis of satellite-derived chlorophyll concentration to the ocean area for which the trajectory concentration was approximately over 0.1% (55–65°S, 40–60ºW for the Weddell Sea and 55–65°S, 60–80°W for the Bellingshausen Sea). DMSP is produced by marine phytoplankton and is the most important precursor of oceanic DMS production. However, the dependence of the oceanic DMS emission on phytoplankton biomass and DMSP concentration is not straightforward owing to the strong variabilities across taxonomic groups and the interplay with environmental factors. Nevertheless, temporal

and spatial distribution of sea-surface DMSP could be an indicator of contemporary DMS emission. In particular, the DMSP-to-chlorophyll ratio could represent the potential DMS production capacity of the ocean because the phytoplankton species with higher cellular DMSP content (i.e., higher DMSP-to-chlorophyll ratio) mostly possess an enzyme that can convert cellular DMSP into DMS, whereas phytoplankton species containing lower DMSP content (i.e., lower DMSP-to-chlorophyll ratio) do not have a DMSP cleavage enzyme (Stefels et al., 2007, Park et al., 2014b and 2018). The total DMSP concentration in the sea-surface was estimated using the algorithm developed by Galí et al. (2015). The algorithm for the total DMSP concentration was based on the satellite-derived chlorophyll concentration and the light exposure regime (see Supplement for more information). We estimated the taxonomic phytoplankton composition of the two ocean domains using the PHYSAT method. This method is a bio-optic model that was specifically developed to identify the dominant phytoplankton groups from ocean color measurements. Phytoplankton groups are generally characterized by specific pigment, shape and size that have different light scattering and absorption properties (Alvain et al., 2005). The PHYSAT method was first developed in 2005 and was used to classify sea-surface phytoplankton into four groups: diatoms, Prochlorococcus, nanoeucaryotes, and Synechococcus (Alvain et al., 2005). Subsequently, the modified PHYSAT method, which can estimate the contribution of the phaeocystis group, was reported in 2008 (Alvain et al., 2008). The PHYSAT method was developed and calibrated based on global data obtained from the sea-viewing wide field-of-view sensor (SeaWiFS) operated from September 1997 to December 2010. In this study, the monthly dataset of five phytoplankton groups at a resolution of 9 km was obtained from the PHYSAT database (http://log.univ-littoral.fr/Physat) estimated using climatology over the SeaWiFS period (1997–2010). Note that "dominant" has been defined as situations in which a given phytoplankton group is a major contributor to the total pigment in a given 9 km resolution (Alvain et al., 2005 and 2008).

**3 Results and Discussion**

**3.1. Seasonal variabilities of nanoparticles at King Sejong Station**

The number concentration of aerosol particles increased gradually from early spring, peaked in the austral summer period (December, January and February) and then began to decrease (Fig. 1a). The number concentration of nanoparticles (2.5–10 nm in diameter, $CN_{2.5-10}$), which is an indication of newly formed particles, also shows distinct seasonal variation (Fig. 1b). A detailed explanation of physical characteristics of new particle formation events (e.g., frequency, formation rate, and growth rate) at King Sejong Station during the same period is explained in Kim et al. (2018). The observation site is surrounded by ocean, sea-ice, and land domains, which may influence new particle formation in different ways. The 2-day air mass back trajectory combined with geographical information showed that approximately 66% of the hourly trajectory points were assigned to the ocean, followed by sea-ice (29%) and land (6%) during the entire study period (Fig. 1c and Fig. S1a). The percentage of hourly trajectory points that passed over the ocean domain were at its maximum during the summer period (79%) when the extent of sea-ice was at its minimum (Fig. S1b). Kim et al. (2018) reported that a total of 101 days were defined as new particle formation events during the eight years and 80 days of new particle formation events occurred when the air mass originated from the ocean domain. Furthermore, 16 days of new particle formation events were observed for the air masses originating from the Antarctic Peninsula. The

remaining five days of events were considered as those of South American origin (three events) and undefined (two events) (see Kim et al. 2018 for the detailed definition and categorization of new particle formation events). The relationship between new particle formation and environmental parameters is complicated, owing to the interplay among multiple sources and complicated processes. The number concentration of the nanoparticles was at its maximum during the productive summer period, and the frequency of new particle formation was the highest when the air mass originated from the ocean domain. Therefore, we focused on the influence of marine biota on the formation of nanoparticles. The hourly mean concentration of nanoparticles matched with the hourly air mass back trajectory in this study. A total of 22,469 hourly mean number concentrations of nanoparticles were measured above the Antarctic atmosphere over the eight years. Approximately 38.2% of the hourly mean number concentration of nanoparticles, which satisfy the >90% retention of hourly trajectory points over the two ocean domains, were used to estimate the link between new particle formation and the oceanic biological characteristics around the observation site. The remaining 61.8% of the hourly mean number concentration of nanoparticles, which do not satisfy the >90% retention over the two ocean domains, were excluded from further analysis. Interestingly, the monthly mean number concentration of nanoparticles that originated from the Bellingshausen Sea was highest in January ($836 \pm 2673$ cm$^{-3}$) and ~2.5-times greater than that which originated from the Weddell Sea ($332 \pm 921$ cm$^{-3}$; Fig. 2b and Table S1). The differences in the number concentration of nanoparticles that originated from the two ocean domains were particularly noticeable during the austral summer period ($568 \pm 249$ cm$^{-3}$ for the Bellingshausen Sea and $262 \pm 66$ cm$^{-3}$ for the Weddell Sea). However, the differences were not evident between March and November ($97 \pm 51$ cm$^{-3}$ for the Bellingshausen Sea and $73 \pm 57$ cm$^{-3}$ for the Weddell Sea; Fig. 2b).

**3.2. Biological characteristics surrounding the observation site**

In general, the abundance and composition of phytoplankton show distinct spatial and seasonal variation in the Antarctic Ocean (Sullivan et al., 1993). Primary production in the Antarctic Ocean is strongly controlled by various factors such as iron limitation, light availability and mixed layer depth (Arrigo et al., 1999; Sedwick et al., 2007; Park et al., 2013a). The composition of the phytoplankton community is poorly studied in the Antarctic Ocean except for the marginal zone at the Antarctic Peninsula. Nevertheless, both phaeocystis and diatoms (mainly *Phaeocystis antarctica* and *Fragilariopsis cylindrus*) are well known as dominant phytoplankton groups in the Antarctic Ocean during the phytoplankton bloom period (Kropuenske et al., 2009; Arrigo et al., 2010). Both diatom and phaeocystis, considered the most ecologically important phytoplanktonic groups, contribute >20% of the global primary productivity, and are particularly abundant at high latitudes (Schoemann et al., 2005; Malviya et al., 2016). The monthly mean chlorophyll concentration around the observation site (55ºS–65ºS, 40ºW–80ºW) began to increase in October and reached its maximum in November and December during the study period (Fig. S2a). The biological characteristics of the two ocean domains showed notable differences. The monthly mean chlorophyll concentration in the Weddell Sea ($0.49 \pm 0.07$ mg m$^{-3}$) was ~70% higher than that of the Bellingshausen Sea ($0.29 \pm 0.05$ mg m$^{-3}$) during the phytoplankton growth period (October–February; Fig. 3a and Fig. S2b). The PHYSAT analysis, which was estimated using the SeaWiFS climatology map, revealed that the distribution of the dominant phytoplankton groups showed distinct patterns (Fig. 3c). Diatoms were the most abundant, and the dominance of the diatom was ~35% during the austral summer period in the Weddell Sea,

followed by nanoeucaryotes (20%), phaeocystis (17%), Prochlorococcus (15%), and Synechococcus (14%) (Fig. 3c and Fig. S3b). Conversely, the dominance of phaeocystis increased significantly and accounted for more than 50% in the Bellingshausen Sea during the austral summer period, while the contribution of the diatom decreased below 10% (Fig. 3c and Fig. S3a). Although the period considered for the SeaWiFS archive (from September 1997 to December 2010) did not coincide with the period of aerosol particle observation at the King Sejong Station (from March 2009 to December 2016), the PHYSAT analysis performed using the SeaWiFS climatology map was successfully applied to the Southern Ocean and could represent the general seasonal trend of the taxonomic composition of marine phytoplankton in the study area (Alvain and d'Ovidio, 2014, Mustapha et al., 2014). Recently, a regional PHYSAT algorithm for the Mediterranean Sea was developed by applying linear interpolation between SeaWiFS and MODIS wavelengths and reflectance threshold (Navarro et al., 2014). However, challenges remain in high-latitude areas such as the Southern Ocean, especially because of the rather sparse matchup available for the calibration and validation of the PHYSAT algorithm (Alvain et al., 2014).

Enzymatic cleavage of planktonic DMSP into DMS is the major source of DMS and the production of DMS and DMSP is species-specific. For example, diatoms, picoplankton (i.e., Synechococcus and Prochlorococcus) and most nanoeucaryotes are known to be DMS(P)-poor species. Conversely, phaeocystis and dinoflagellates have a high cellular DMSP content and many of them possess a DMSP cleavage enzyme that can convert DMSP into gaseous DMS (Keller, 1989; Stefels et al., 2007; Park et al., 2014b). The conversion of cellular DMSP into DMS is controlled by not only the concentration of DMSP but also, more importantly, by the DMSP cleavage enzyme. DMS is often produced following the local chlorophyll maxima, leading to a lag period (several weeks to months) (Polimene et al., 2012). This phenomenon is evident when the concentration of DMSP is largely contributed by DMSP-poor species such as diatoms. Most DMSP-poor species do not possess a DMSP cleavage enzyme, and therefore, the conversion of DMSP into DMS occurs when the cellular DMSP is released into the ocean as a form of dissolved matter. Subsequently, some dissolved DMSP degrades into gaseous DMS through the bacterial DMSP enzymatic cleavage (Simó, 2001; Stefels et al., 2007). However, a larger proportion of dissolved DMSP is assimilated into bacterial tissues through demethylation processes, which do not produce gaseous DMS (Todd et al., 2007; Reisch et al., 2011). A direct correlation between the local chlorophyll concentration and atmospheric DMS mixing ratio in the absence of lag periods was observed in the Arctic Ocean where *Phaeocystis pouchetii* (containing both high cellular DMSP and DMSP cleavage enzyme) dominates (Park et al., 2013b). Moreover, the DMS production capacity in the Arctic Ocean was more significantly controlled by the abundance of DMSP-rich phytoplankton than the total biomass of phytoplankton (Park et al., 2018). These results indicate that the blooming of phytoplankton species containing higher cellular DMSP content results in a much higher DMS production capacity than the blooming of DMSP-poor phytoplankton species. Therefore, the DMSP-to-chlorophyll ratio is commonly used to explain the differences in taxonomic compositions affecting the oceanic DMS-production capacity (e.g., Belviso et al., 2000; Stefels et al., 2007; Tison et al., 2010; Park et al., 2014b and 2018). In particular, *Phaeocystis antarctica* was reported to be a dominant species in terms of DMS production in the Antarctic Ocean during the bloom period (Gibson et al., 1990; Schoemann et al., 2005), exhibiting a cellular DMSP concentration in phaeocystis several times that of diatoms (Hatton and Wilson, 2007; Stefels et al., 2007). The sea-surface DMSP concentration surrounding the observation site was estimated using a newly developed algorithm and was 30% higher in the Weddell Sea than in the Bellingshausen Sea during the summer period, possibly owing to intense blooming of DMSP-containing diatoms in the Weddell Sea (Fig. 3b and Fig S4a). This could illustrate that, despite

having lower cellular DMSP content than phaeocystis, diatoms dominated the overall DMSP production in the Weddell Sea owing to their much larger biomass. However, the DMSP-to-chlorophyll ratio in the Bellingshausen Sea ($110.2 \pm 27.8$ mmol g$^{-1}$) was ~2-fold higher than that of the Weddell Sea ($72.2 \pm 8.3$ mmol g$^{-1}$) between December and February in 2009–2016 (Fig. 3d and Fig. S4b), possibly owing to the relatively higher contribution of the DMSP-rich phaeocystis group in the Bellingshausen Sea.

### 3.3. Influence of phytoplankton on aerosol formation

Biogenic trace gases produced by marine phytoplankton (i.e., DMS, isoprene, and halogenated gases) are known to be the key compounds contributing to the formation of new particles in the remote marine environment; however, quantifying the relationship between new particle formation events and marine biology is a major challenge (Brooks and Thornton, 2018). MSA in the marine atmosphere forms exclusively from the photooxidation of DMS, and shows strong seasonal variation (Ayers and Gras, 1991; Savoie et al., 1993; Preunkert et al., 2008). A previous study has reported that the highest values for both MSA and the scattering Ångström exponent (SAE; qualitative examination of the aerosol optical mean size) were observed at the Marambio Station (64.3°S, 56.6°W) on the Antarctic Peninsula during austral summer in 2013–2015 (Asmi et al., 2018). The MSA concentration of the fine aerosol particles measured at the King Sejong Station during the summer period in 2013 and 2014 was broadly consistent with the number concentration of nanoparticles. The MSA concentration shows distinct daily variations. The mean MSA concentration was $72.6 \pm 99.1$ ng m$^{-3}$ (ranged from 4.2 to 657.0 ng m$^{-3}$) (Fig. 4a), similar to the values observed at six Antarctic sites during the productive summer period (e.g., Prospero et al., 1991; Minikin et al., 1998; Preunkert et al., 2007; Read et al., 2008; Zhang et al., 2015; Asmi et al., 2018). To identify the potential origin of MSA, air mass back trajectories were determined and the retention time above each domain was averaged for the corresponding 24 h sampling time. When applying 3-, 4-, and 5-day air mass back trajectories, the number of samples that satisfy >90% retention in the Bellingshausen and Weddell Seas was less than 20% of the total MSA samples owing to its longer transport pathway. Inevitably, the air mass origin of MSA was divided into two sectors i.e., the Bellingshausen Sea sector (<58.8°W) and the Weddell Sea sector (>58.8°W) by selecting the air mass back trajectories with >50% retention in a given sector. Although the limited number of samples of MSA (84 samples at daily intervals) collected during the summer periods in 2013 and 2014 may not be sufficient to identify its source origin exactly, the MSA concentration also showed distinct differences depending on the air mass origin. The inflow of the air masses from the Bellingshausen Sea increased the concentration of MSA in the aerosol particles. Notably, the MSA concentration that originated from the Bellingshausen Sea sector ($87.6 \pm 110.0$, $86.6 \pm 110.0$, and $83.9 \pm 109.0$ ng m$^{-3}$ for 3-, 4-, and 5-day air mass back trajectories based estimates, respectively) was ~3-times higher than that which originated from the Weddell Sea sector ($27.4 \pm 19.3$, $30.6 \pm 27.5$, and $33.9 \pm 30.4$ ng m$^{-3}$ for 3-, 4-, and 5-day air mass back trajectories based estimates, respectively) during the austral summer period in 2013–2014 (Fig. 4b). Although the period of satellite observations and in situ chemistry analysis is not exactly the same, both satellite-derived biological characteristics and aerosol chemistry data support the interpretation that there was higher abundance of DMS(P)-rich phytoplankton in the Bellingshausen Sea than in the Weddell Sea during the austral summer period (Fig. 3, Fig. 4, Figs. S2, S3 and S4).

In the 8-year record, the monthly mean chlorophyll concentration was positively correlated with the monthly mean number concentration of nanoparticles for the air masses that originated from the Bellingshausen Sea in January and February ($r^2 = 0.69$, $n = 12$, $P < 0.05$; Fig. 5a). During this period, the contribution of the DMS(P)-rich phaeocystis to the chlorophyll concentration was highest in the Bellingshausen Sea (i.e., DMSP-to-chlorophyll ratio >100 mmol g$^{-1}$; dominance of phaeocystis >50%; dominance of diatoms <10%; Fig. 3, Fig. S3a and Fig. S4). Conversely, the increase in the chlorophyll concentration was not correlated with the increase in the number concentration of nanoparticles in the Weddell Sea (Fig. 5b). As a consequence, the higher occurrence of nanoparticles from the Bellingshausen Sea inferred from our analysis was likely to be associated with a higher abundance of DMS(P)-rich phytoplankton, whereas the lower occurrence of nanoparticles from the Weddell Sea appeared to be associated with a higher abundance of DMS(P)-poor phytoplankton.

**4 Conclusions**

The physical properties of aerosol particles measured above the remote Antarctic Peninsula over 8 years were analyzed in conjunction with the satellite-derived biological characteristics around the observation site. These results show that the formation of nanoparticles was strongly associated not only with the biomass of phytoplankton but, more importantly, also its taxonomic composition in the Antarctic Ocean. Previous studies have reported that diatoms have a competitive advantage under conditions in which the mixed layers are shallow and the light levels are relatively high. Conversely, phaeocystis is well adapted to conditions in which mixed layers are deep and light levels are variable (e.g., Weber and El-Sayed, 1987; Arrigo et al., 1999; Goffart et al., 2000; Alvain et al., 2008; Arrigo et al., 2010). These results are consistent with the distribution of phytoplankton groups in the Bellingshausen and Weddell seas. Given that the mixed layer depth in the Bellingshausen Sea (45.6 ± 4.1 m) was relatively deeper than that of the Weddell Sea (36.2 ± 3.8 m; Fig. S6) during the austral summer period, the growth of DMS(P)-rich phaeocystis may therefore be more favorable in the Bellingshausen Sea. Sea-surface warming and freshening is commonly associated with a shallowing of the mixed layer depth (Capotondi et al., 2012). The warming trend has shown the spatial complexity across the Antarctic Ocean in recent decades (Turner et al., 2005). Therefore, all regions of the Antarctic Ocean will experience different changes in phytoplankton productivity and taxonomic composition in response to the climate change (Deppeler and Davidson, 2017).

In this study, we have focused on the relationship between the formation of nanoparticles and marine biota. The formation of secondary aerosols contributes significantly to the atmospheric aerosol number and accounts for 30–80% of the global cloud condensation nuclei (Merikanto et al., 2009; Westervelt et al., 2014; Sanchez et al., 2018; Sullivan et al., 2018). Our results indicate that changes in the taxonomic composition of marine phytoplankton (i.e., DMS(P)-rich species vs. DMS(P)-poor species) could have a significant impact on the aerosol properties in the remote marine environment. Precursors other than biogenic DMS could play a key role in the formation of new particles in the Antarctic atmosphere. In fact, 16 days of new particle formation events out of 101 events were observed when the air mass originated from the Antarctic Peninsula during the study period. Penguin colonies are dispersed throughout the Antarctic Peninsula (Croxall et al., 2002), and the emission of ammonia

from these colonies could trigger the formation of nanoparticles (Weber et al., 1998; Croft et al., 2016). Moreover, iodine molecules produced by biotic and abiotic processes near sea-ice region are known to influence the formation of aerosol particles (Allan et al., 2015; Sipilä et al., 2016). Future studies are required to minimize the knowledge gaps related to multiple precursors and their source origins. Specifically, continuous measurements of

the physiochemical properties of aerosol particles and molecular-scale measurements of chemical species (e.g., sulfur-, nitrogen-, and halogen-containing compounds) involved in nucleation processes are required to provide direct evidence for the contribution of these compounds to the formation and growth of aerosol particles and to understand their climate feedback roles in the remote marine environment.

**Author contributions**

KTP, JEH, JSK, TWK and YYJ designed the study. YYJ and YTG analyzed the physical properties of the aerosol particles. HSB, SB and RT operated the air sampler and analyzed the MSA. KTP and JEH wrote the manuscript.

**Acknowledgements**

This study was supported by the KOPRI project (PE19010, PE19140). We thank overwintering staff for assisting us in maintaining the aerosol equipment at the King Sejong Station.

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

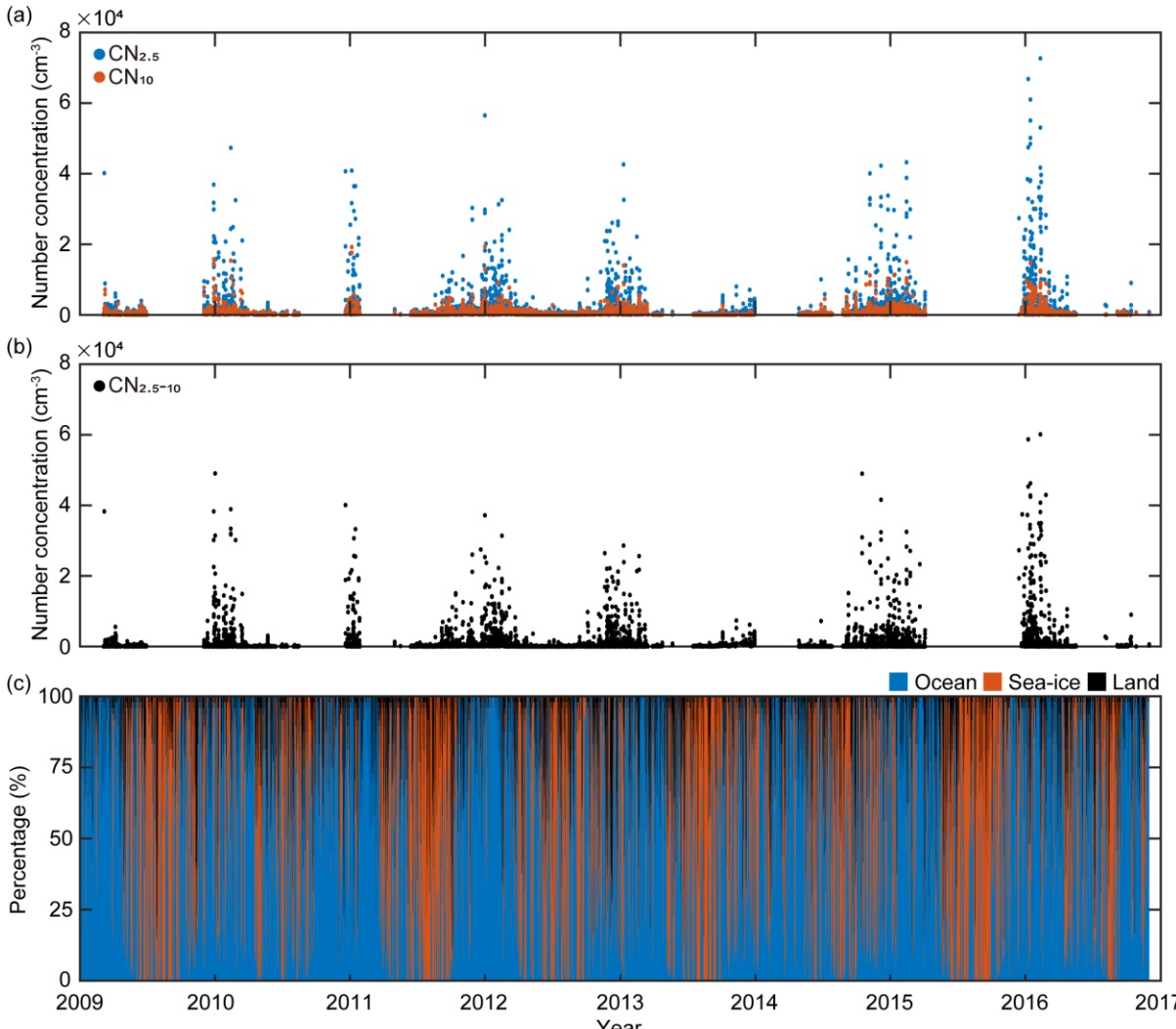

**Figure 1:** (a) Hourly variations in the number concentration of particles >2.5 nm in diameter ($CN_{2.5}$, blue symbols) and the number concentration of particles >10 nm in diameter ($CN_{10}$, red symbols), (b) hourly variations in the number concentration of nanoparticles (ranging from 2.5 to 10 nm in diameter) calculated using the differences between $CN_{2.5}$ and $CN_{10}$, and (c) hourly variations in the retention time of 2-day air mass back trajectories over the three domains including ocean (blue), sea-ice (red), and land (black) domains from March 2009 to November 2016.

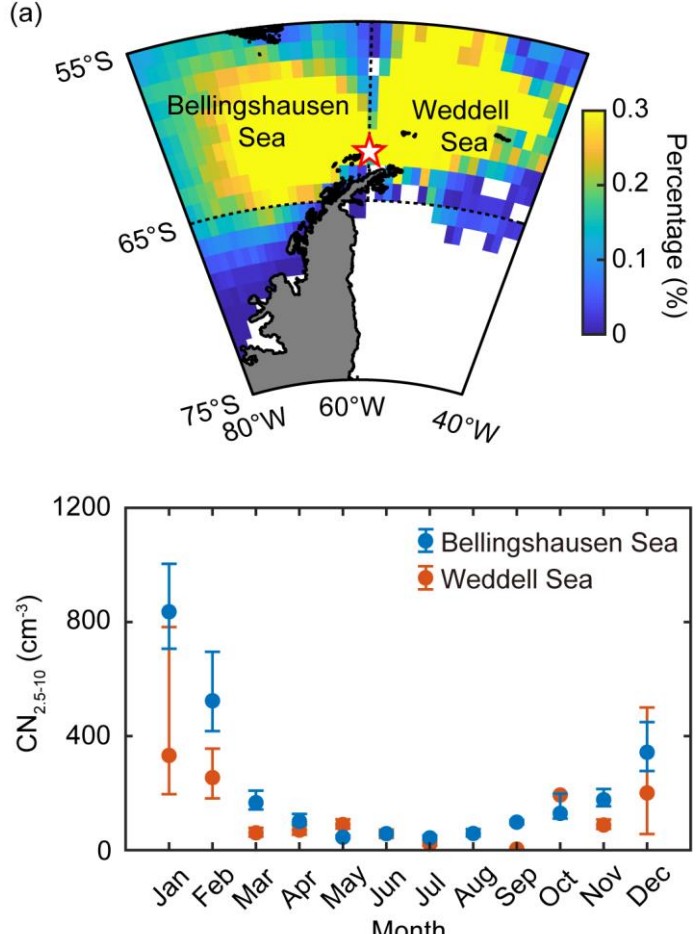

**Figure 2:** (a) Back-trajectories of the air masses arriving at King Sejong Station (62.2ºS, 58.8ºW; star symbol), Antarctic Peninsula. The colors indicate the percentage (%) of the air mass located at that spot during the 2 days prior to arriving at the observation site. Note that the air mass back-trajectories that did not have >90% retention in the two selected ocean domains (i.e., Bellingshausen and Weddell Seas) were excluded. (b) Seasonal variation of nanoparticles (2.5–10 nm in diameter, $CN_{2.5-10}$) observed at King Sejong Station between March 2009 and December 2016. Blue and red symbols indicate the number concentration of nanoparticles that originated from the Bellingshausen and Weddell seas, respectively. The error bars indicate the 95% confidence interval estimated by bootstrap method from the monthly $CN_{2.5-10}$ data.

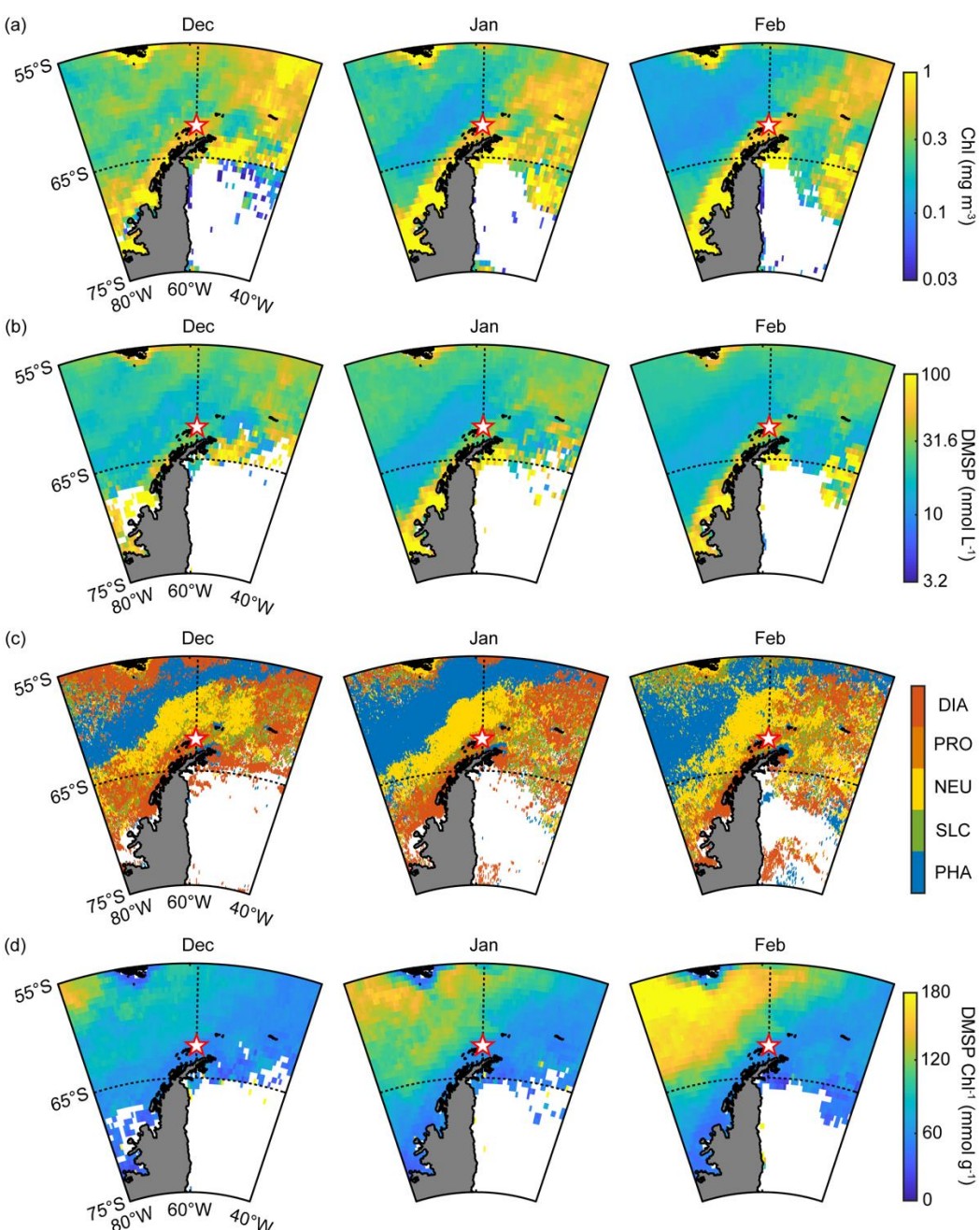

Figure 3: (a) Monthly mean chlorophyll concentration during the months of December, January, and February in 2009–2016, (b) monthly mean DMSP concentration during the months of December, January, and February in 2009–2016, (c) phytoplankton taxonomic composition including diatoms (DIA), Prochlorococcus (PRO), nanoeucaryotes (NEU), Synechococcus (SLC), and phaeocystis (PHA) estimated using the PHYSAT method with the climatology map obtained from SeaWiFS archive, and (d) monthly mean DMSP-to-chlorophyll ratio during the months of December, January, and February in 2009–2016.

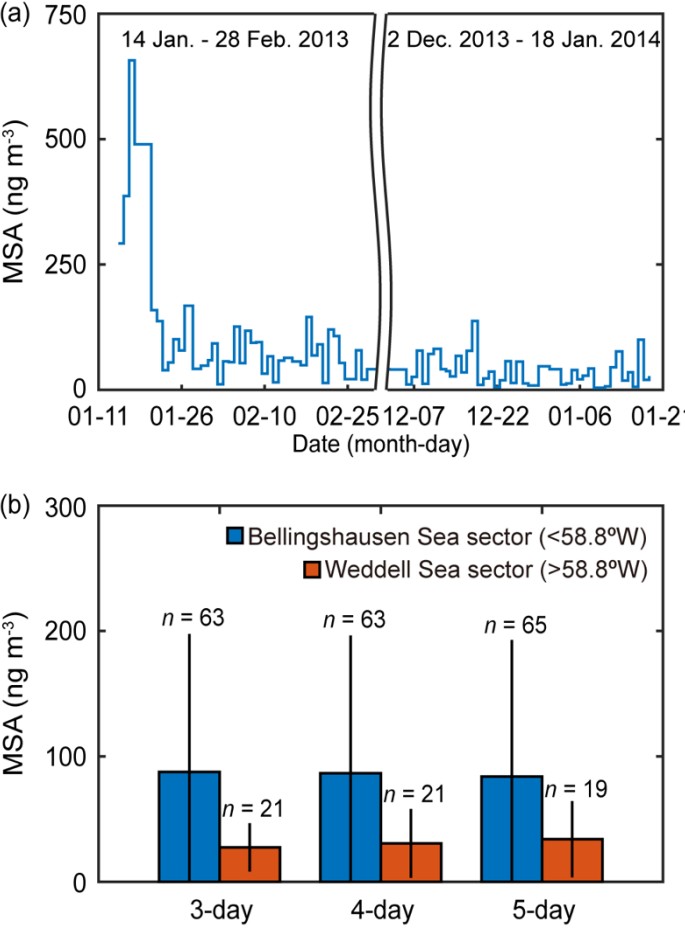

Figure 4: (a) Daily concentration of MSA collected at the sampling site during the summer periods in 2013 and 2014 (explicitly, from 14 January to 28 February in 2013, and from 2 December 2013 to 18 January 2014). (b) The mean MSA concentration that potentially originated from the Bellingshausen Sea sector (<58.8°W) and the Weddell Sea sector (>58.8°W) estimated by applying 3-, 4-, and 5-day air mass back trajectories during the austral summer periods in 2013 and 2014. The error bars indicate 1 standard deviation (1σ) from the mean values.

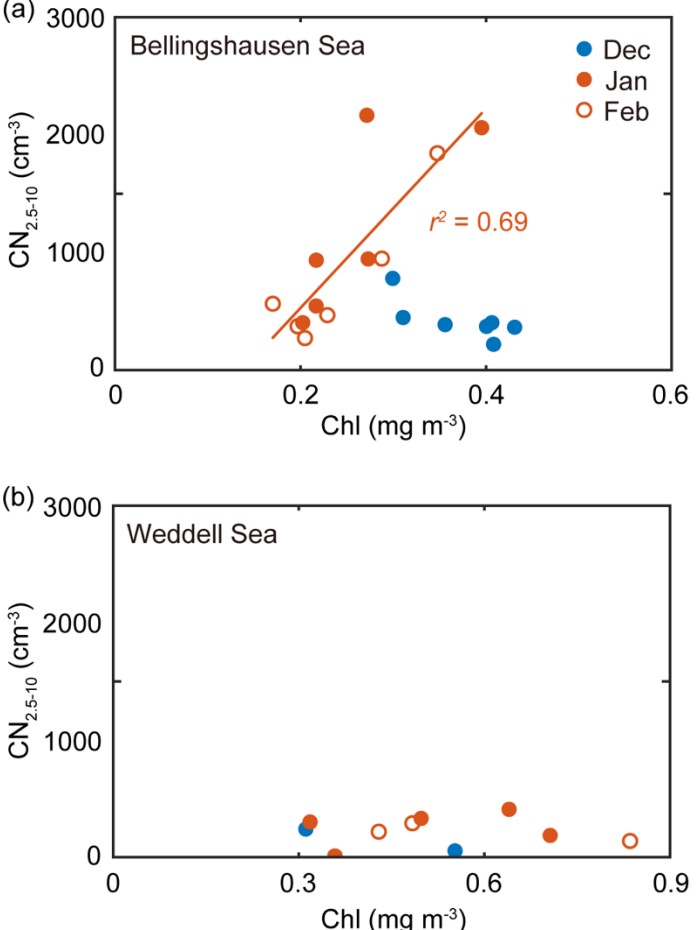

Figure 5: (a) Relationship between the monthly mean chlorophyll concentration for the Bellingshausen Sea (55°S–65°S, 60°W–80°W) and the monthly mean number concentration of nanoparticles that originated from the Bellingshausen Sea in 2009–2016. (b) Relationship between the monthly mean chlorophyll concentration for the Weddell Sea (55°S–65°S, 40°W–60°W) and the monthly mean number concentration of nanoparticles that originated from the Weddell Sea in 2009–2016. The filled blue, filled red and open red symbols indicate the data obtained in December, January and February, respectively. The solid lines represent the best fit.

