# Peer review of "New particle formation events observed at the King Sejong Station, Antarctic Peninsula – Part 2: Link with the oceanic biological activities"

_Atmospheric Chemistry and Physics, 2018_

## Referee Comment (RC1) · Anonymous Referee #1 · 18 Dec 2018

General comments: This manuscript (ms) is the second part on new particle formation (NPF) events observed at the Korean Antarctic Station King Sejong. The objective of this remarkably brief accompanying study is to identify the interrelation between NPF and biological activity as well as phytoplankton taxonomic composition in the most dominant source region for this site, the Weddell and Bellinghausen Sea. The authors present authentic and original scientific material that potentially has relevant implications for understanding atmospheric processes in Antarctica and is an important contribution on this field of research. Hence, the subject is appropriate to ACP and I recommend accepting the paper after some (major) revisions. My principal concern is that the main conclusion of this study, i.e. the link between phytoplankton taxonomic

composition and NPF is not convincingly presented and the scientific approach is not clearly outlined (see my comments below).

Specific comments:

1. The discussion presented in Chapters 3.2 and 3.3 is not conclusive. I am confused about the described approach deriving DMSP concentrations and DMSP/Chl ratios. The Galí et al. algorithm presented in the Supplement tells me, that DMSP concentrations are dependent on Chl concentrations, SST, MLD and PIC. What is the impact of SST, MLD and PIC compared to Chl? Did you resort to the phytoplankton taxonomic composition derived from the PHYSAT database to weight the results? In this case: How representative is the PHYSAT database for the period 1997-2010 with regard to your observation period?

2. However, even agreeing that DMSP/Chl ratios were (generally) higher in the Bellinghausen Sea area, from this fact alone, you cannot conclude that DMS production was higher. For this purpose, one has at least to compare the absolute DMSP concentrations, not just the ratios as presented in Fig. 2c. Mean chlorophyll concentrations are much higher in the Weddell Sea; hence, a systematically lower ratio in this region is just comprehensible, even in the case of DMSP concentrations being higher compared to the Bellinghausen Sea.

3. In addition, it is not clear in which way you generate the pixels shown in Fig. 2b: What is the threshold value for a given phytoplankton contingent to be depicted as dominant? In the same way, the assertion on page 5 line 36-37 is equivocal: "...35% of the satellite pixels were dominated by diatoms...". What does this mean and what is about the remaining 65%?

Chapter 3.1, line 12-14: About 38.2% of the hourly mean number concentration of nanoparticles complied with the >90% criterion. Did the remaining 61.8% of the data indicate any significant link to the origin of the air masses?

Chapter 3.3, first passage and Fig. 3: MSA data are only available for one summer season (2013/14). Due to the fact, that MSA concentrations around Antarctica have proven to be extremely variable on every timescale (see e.g. Minikin et al. 1998), the representativeness of the data may be questionable. How many individual filter measurements are represented by each bar shown in Fig. 3? It would be much more informative presenting here the original (daily) MSA concentration data.

Technical correction:

Page 3, line 34: "...below the detection" should be "...below the detection limit."

---

## Short Comment (SC1) · 21 Dec 2018

This paper represents some interesting observations on what is clearly an important atmospheric phenomenon. However, I feel I should comment on the authors' working hypothesis, which is that DMS emissions are responsible for the new particle formation (NPF) observed. While this is undoubtedly a major source of particulates in the marine environment, there has recently been an amount of evidence in support of iodine compounds also being responsible for NPF in Arctic and Antarctic regions (see references below). It would be highly informative to gain the authors' view of this, specifically whether abiotic or biotic iodine releases from the ice or ocean could also

be contributing to the correlation noted here.

Allan, J. D., Williams, P. I., Najera, J., Whitehead, J. D., Flynn, M. J., Taylor, J. W., Liu, D., Darbyshire, E., Carpenter, L. J., Chance, R., Andrews, S. J., Hackenberg, S. C., and McFiggans, G.: Iodine observed in new particle formation events in the Arctic atmosphere during ACCACIA, Atmos. Chem. Phys., 15, 5599-5609, 10.5194/acp-15-5599-2015, 2015.

Atkinson, H. M., Huang, R. J., Chance, R., Roscoe, H. K., Hughes, C., Davison, B., Schönhardt, A., Mahajan, A. S., Saiz-Lopez, A., Hoffmann, T., and Liss, P. S.: Iodine emissions from the sea ice of the Weddell Sea, Atmos. Chem. Phys., 12, 11229-11244, 10.5194/acp-12-11229-2012, 2012.

Dall′Osto, M., Simo, R., Harrison, R. M., Beddows, D. C. S., Saiz-Lopez, A., Lange, R., Skov, H., Nøjgaard, J. K., Nielsen, I. E., and Massling, A.: Abiotic and biotic sources influencing spring new particle formation in North East Greenland, Atmos. Environ., 190, 126-134, 10.1016/j.atmosenv.2018.07.019, 2018.

Raso, A. R. W., Custard, K. D., May, N. W., Tanner, D., Newburn, M. K., Walker, L., Moore, R. J., Huey, L. G., Alexander, L., Shepson, P. B., and Pratt, K. A.: Active molecular iodine photochemistry in the Arctic, P Natl Acad Sci, 114, 10053-10058, 10.1073/pnas.1702803114, 2017.

Roscoe, H. K., Jones, A. E., Brough, N., Weller, R., Saiz-Lopez, A., Mahajan, A. S., Schoenhardt, A., Burrows, J. P., and Fleming, Z. L.: Particles and iodine compounds in coastal Antarctica, J. Geophys. Res.-Atmos., 120, 7144-7156, 10.1002/2015JD023301, 2015.

Sipilä, M., Sarnela, N., Jokinen, T., Henschel, H., Junninen, H., Kontkanen, J., Richters, S., Kangasluoma, J., Franchin, A., Peräkylä, O., Rissanen, M. P., Ehn, M., Vehkamäki, H., Kurten, T., Berndt, T., Petäjä, T., Worsnop, D., Ceburnis, D., Kerminen, V.-M., Kulmala, M., and O'Dowd, C.: Molecular-scale evidence of aerosol particle formation via

sequential addition of HIO3, Nature, 537, 532, 10.1038/nature19314, 2016.

---

## Referee Comment (RC2) · Anonymous Referee #2 · 4 Jan 2019

This is a very compact manuscript dealing with a very important topic. The paper appears original, but the authors need to demonstrate it more clearly in the paper. I have also a few other scientific issues to be answered and revised before the paper can be accepted for publication.

Scientific issues

The authors state that they aim to find a link between atmospheric new particle formation (NPF) and marine biota. I think that this paper succeeds in doing that. However, I would like to see more comprehensive discussion on how this result build upon earlier findings by other researchers and what exactly is the new scientific message that this

paper brings into this important topic.

The discussion about atmospheric NPF mechanisms in the Introduction is based on studies published more than 10 years ago. Much new has been learned on this topic since then. The paper would benefit from updating this discussion in line with most recent published work.

The authors use sub-10 nm particles to identify recent atmospheric NPF and bulk aerosol samples to get idea on how much particulate MSA has been formed in measured air masses. Then, 2-day air mass back trajectories are used to identify the origin of measured air. This is fine when looking at NPF, but how about bulk aerosol samples? Typical lifetime of PM mass is a few days in the lower troposphere, so 2-day trajectories may not tell the whole story about the origin of this MSA. The authors should mention this and perhaps comment it in the paper.

Minor scientific and technical issues

Please be careful with the term "aerosols" – in many cases "aerosol particles" would be more appropriate. Also, naming particles formed by NPF via pathways involving organic compounds as secondary organic aerosols is a bit confusing. This is because the term "secondary organic aerosol" is generally used for describing particulate matter (in terms of mass) formed in the atmosphere, not just newly-formed particles.

page 5, line 7: this submitted?

page 6, lines 31-33 and Fig. 4a: Why there are only 6 measurement points in the figure, although the measurements cover 8 years?

Figure 2c and the text: I am a bit confused on the meaning of DMSP to chlorophyll ratio. If DMSP concentration is the relevant quantity to look at, then why to scale it with chlorophyll? Or is there any scaling? The unit in Fig. 2c refers to concentration, not to any ratio between air and sea-water concentrations.

Fig. 3: in x-axis there should be DEC, not DEB

[Figure]

---

## Referee Comment (RC3) · Anonymous Referee #3 · 1 Feb 2019

The current paper of Jang et al (2018) shows some interesting data from the King Sejong Station in Antarctica. Whilst the data are of great interest (although some already published in the same journal last year), this is an additional analysis. However, the paper in its current form is not suitable for publication, I reject this paper but I suggest a resubmission in a lower impact journal, or a more in depth analysis before considering it suitable for ACP.

Specifically, only 38% of the data are analysed, and conclusions are made but not well supported in its current form. I suggest to analyse the whole dataset and to present a whole analysis, including air masses arriving from other sectors, including península,

and weddell sea and all the South regions not currently included.

As regards of the filter data, there are only few months of data presented, this should be stated in the abstract, so the reader know what type of analysis is carried out, this is not a long term monitoring as it is stated.

Two points should be also discussed. 100 ng m3 may be too high (usually it is used 10, 20, max 40 ng m-3) and local pinguin emissions near the base should also be discussed.

The literatura review is also not very complete, the current issues debated in the literatura now well discussed and summarised (pinguing emissions, marginal sea ice emissions, chemical involved in the emissions as pointed out in the comment by Dr. J Allan).

In summary, the authors need to carry on a more in depth analysis with the whole dataset avaiable, and in particular the 100% of the air masses coming from all sectors, so the reader can really undertand the whole oceanic activities, and not a very minor part presented.
* * *

---

## Author Comment (AC1) · 10 Apr 2019

We thank Referee 1 for providing insightful suggestions that have considerably improved the readability of the revised manuscript. Our responses to general and specific comments raised by Referee 1 are stated below. The revised manuscript was uploaded in the form of a supplement.

General comments

The link between phytoplankton taxonomic composition and NPF is not convincingly presented and the scientific approach is not clearly outlined: We have added a de-

scription and figures to support our key findings in the revised manuscript. In short, the explanation regarding the biological characteristic of the surrounding ocean has been thoroughly complemented in the revised manuscript. Figures for 8-year CN records (CN2.5, CN10, and CN2.5-10), 8-year transport history at an hourly interval, sea-surface DMSP concentration, and daily MSA concentration have been newly added. We have addressed these issues in detail in our response to specific comments.

Specific Comments

1. The discussion presented in Chapters 3 is not conclusive, in particular, the description regarding DMSP-to-chlorophyll ratio is confusing (Issue 1). What is the impact of SST, MLD and PIC compared to Chl (Issue 2)? How representative is the PHYSAT database for the period 1997–2010 with regard to your observation period? (Issue 3):

(Response to Issue 1) We have thoroughly revised the "Results and Discussion (chapter 3)" part to improve the readability of our manuscript. First, we have added a paragraph explaining the DMSP-relevant processes and their association with marine biota to clarify the meaning of the DMSP-to-chlorophyll ratio (P4, line 34 – P5, line 6; P7, line 16 – P8, line 1; P8, lines 7–10). In short, the conversion of cellular DMSP into DMS is controlled by not only the concentration of DMSP but also, more importantly, the DMSP cleavage enzyme. The phytoplankton species containing high cellular DMSP (i.e., high DMSP-to-chlorophyll ratio) mostly possess an enzyme that can convert cellular DMSP into DMS, whereas phytoplankton species containing low DMSP content (i.e., low DMSP-to-chlorophyll ratio) do not have a DMSP cleavage enzyme. Therefore, the DMSP-to-chlorophyll ratio is commonly used to explain the differences in taxonomic compositions affecting the oceanic DMS-production capacity (e.g., Belviso et al., 2000; Stefels et al., 2007; Tison et al., 2010; Park et al., 2014b and 2018). Explanation regarding the PHYSAT analysis and MSA has also been modified in the revised manuscript (see authors' response 1-3 and 5 for more details).

(Response to Issue 2) DMSP is produced by marine phytoplankton; however, the

dependence of the production of planktonic DMSP on phytoplankton biomass is not straightforward owing to the strong variabilities across taxonomic groups and the interplay with environmental factors. Gali et al. (2015) developed a DMSP algorithm based on satellite-derived chlorophyll (to measure phytoplankton biomass) and the light exposure regime (to measure key environmental factors controlling DMSP production). The terms SST and MLD have been used to validate the environmental factors controlling DMSP production. Specifically, euphotic layer depth (Zeu) and mixed layer depth (MLD) dataset were applied to establish a mixing state of the sea surface (stratified vs. mixed water column), and the variability in modeled and measured DMSP was improved by adding sea-surface temperature and log10(Zeu/MLD) as predictors for the stratified and mixed subsets in the proposed algorithm. Additionally, a sub-model based on particulate inorganic carbon (PIC) was developed to complement DMSP diagnosis in coccolithophore blooms, where satellite chlorophyll concentration may not be reliable. We have added these sentences in the revised supplementary (P2, lines 2–11).

(Response to Issue 3) The PHYSAT method was developed based on the SeaWiFS dataset, which is available from 1997 to 2010, and is the most widely used algorithm for the estimation of the taxonomic composition of marine phytoplankton. The PHYSAT analysis calculated using the SeaWiFS climatology map was successfully applied to the Southern Ocean and could represent the general seasonal trend of the taxonomic composition of marine phytoplankton in the study area (Alvain and d'Ovidio, 2014). Recently, a regional PHYSAT algorithm for the Mediterranean Sea was developed by applying linear interpolation between SeaWiFS and MODIS wavelengths and reflectance threshold and is available at a global scale (Navarro et al., 2014). However, challenges remain in high-latitude areas such as the Southern Ocean, especially because of the rather sparse matchup available for the calibration and validation of the PHYSAT algorithm (Alvain et al., 2014). We have calculated the taxonomic composition of marine phytoplankton by using the MODIS-based PHYSAT algorithm (see the figure below). In this study area, the MODIS-based PHYSAT algorithm overestimated the dominance of diatoms compared with the SeaWiFS-based PHYSAT algorithm, whereas the dominance of phaeocystis was underestimated. Nevertheless, the PHYSAT results estimated using both the SeaWiFS archive (from January 1997 to December 2010) and MODIS archive (from January 2002 to December 2016) show more than three times higher dominance of the phaeocystis group in the Bellingshausen Sea than in the Weddell Sea during the summer period. In the revised manuscript, we have changed "...obtained PHYSAT database, and was estimated between 1997 and 2010" to "...obtained from the PHYSAT database estimated using climatology over the SeaWiFS period (1997–2010)" (P5, lines 18–19). We also have changed "which was applied from 1997 to 2010" to "which was estimated using the SeaWiFS climatology map" (P6, line 35) for better clarity. An additional paragraph explaining the feasibility of the SeaWiFS-based PHYSAT method has been added in the revised manuscript (P5, lines 12–21; P7, lines 3–11).

2. Need to compare the absolute DMSP concentration, not just the DMSP-to-chlorophyll ratio: As this referee suggested, we have added figures for sea-surface DMSP concentration near the observation site (Fig. 3b and Fig. S4a). A brief explanation regarding the DMSP concentration has also been added (P7, line 36 – P8, line 1). The DMSP concentration was ∼30% higher in the Weddell Sea during the blooming period, possibly owing to intense blooming of DMSP-containing diatoms. This could illustrate that, despite having lower DMSP-to-chlorophyll ratios than phaeocystis, diatoms dominated the overall DMSP production in the Weddell Sea owing to their much larger biomass. However, we cannot fully support this hypothesis owing to the absence of field measurement of taxonomic compositions of phytoplankton in this vast remote ocean.

3. Provide clear description regarding the results for the PHYSAT analysis: The PHYSAT method is a bio-optic model specifically developed to identify the dominant phytoplankton groups. Here, "dominant" has been defined as situations in which a given phytoplankton group is a major contributor to the total pigment at a given 9 km

resolution (Alvain et al., 2005 and 2008). This paragraph has been added to the revised manuscript (P5, lines 19–21). We have replaced "∼35% of the satellite pixels were dominated by diatoms" to "the dominance of the diatom was ∼35% during the austral summer period in the Weddell Sea, followed by nanoeucaryotes (20%), phaeocystis (17%), Prochlorococcus (15%), and Synechococcus (14%)" for better clarity (P6, line 37–38).

4. The statement –"About 38.2% of the hourly mean number concentration of nanoparticles complied with the >90%criterion." Did the remaining 61.8% of the data indicate any significant link to the origin of the air masses? The formation of new particles in the remote marine atmosphere is complicated owing to multiple sources and complicated processes. The remaining 61.8% of hourly mean number concentration of nanoparticles may have undergone more complicated transport history. Therefore, the hourly mean number concentration of nanoparticles, which do not satisfy the >90% retention over the two ocean domains, were excluded from further analysis. Because it was not easy to find a strong relationship between the formation of nanoparticles and the multiple potential source origin (e.g., marine biota, sea-ice extent, and penguin colony). A total of 101 days were defined as new particle formation events during the eight years and 80 days of new particle formation events occurred when the air mass originated from the ocean domain. The number concentration of the nanoparticle was at its maximum during the productive summer period, and the frequency of new particle formation was the highest when the air mass originated from the ocean domain. Therefore, we focused on the influence of marine biota on the formation of nanoparticles in this study. In the revised manuscript, we have added more description to clarify the scope of the present study (P1, line 16; P1, lines 20–22; P5, line 34 – P6, line 6; P6, lines 11–13). Furthermore, the limitations of the present study and the scope of future studies have also been added (P19, line 33 – P10, line 8).

5. How many individual filter measurements are represented by each bar shown in Fig. 3? Provide daily MSA concentration data: 52 individual measurements for MSA

(43 measurements for the Bellingshausen Sea and nine measurements for the Weddell Sea) were used for Fig. 3 in the previous version of the manuscript. As this referee suggested, we have added daily MSA concentration data in the revised manuscript (Fig. 4a). An explanation of MSA variation has also been added (P8, lines 16–27; P8, lines 30–33). We have revised the figure for MSA concentration in response to the query raised by referee #2. In the revised figure, we have applied 3-, 4-, and 5-day back trajectories to analyze the potential origin of MSA, because the retention time of PM10 is known to be >2 days in the low troposphere. The extension of the air mass back trajectory time points altered the percentage of air mass retention above the major domains. The number of samples that satisfy >90% retention in the Bellingshausen and Weddell Seas was less than 20% of the total MSA samples owing to its longer transport pathway. Inevitably, the air mass origin of MSA was divided into two sectors i.e., the Bellingshausen Sea sector (<58.8oW) and the Weddell Sea sector (>58.8oW) by selecting the air mass back trajectories with >50% retention in a given sector. Thus, all the measurements (84 individual samples) were allocated to one of the two sectors. The mean MSA concentration and the number of individual measurements are shown in Fig. 4b.

Technical correction

6. P3, line 34: We have replaced "…below the detection" to "below the detection limit" (P4, line 3).

Please also note the supplement to this comment:
https://www.atmos-chem-phys-discuss.net/acp-2018-1181/acp-2018-1181-AC1-supplement.pdf
* * *
**Fig. 1.** PHYSAT analysis based on (a) SeaWiFS climatology map (1997-2010), and (b) MODIS climatology map (2002-2016)

---

## Author Comment (AC2) · 10 Apr 2019

We thank Referee 2 for providing valuable suggestions that improved the readability of our revised manuscript. Our responses to this Referee's scientific and technical comments are stated below. The revised manuscript was uploaded in the form of a supplement.

Scientific issues

1. Need to provide more comprehensive discussion on how this result build upon earlier findings and what exactly is the new scientific message: The key finding of this study is

that the formation of nanoparticles was strongly associated with not only the biomass of phytoplankton but also, more importantly, its taxonomic composition in the Antarctic Ocean; therefore, changes in the taxonomic composition of marine phytoplankton (i.e., DMS(P)-rich species vs. DMS(P)-poor species) may have a significant impact on the aerosol properties in the remote marine environment. In the revised manuscript, we have added a paragraph explaining earlier finding and the key results of this study. Furthermore, the limitations of the present study and the scope of future studies have also been added (P9, line 29 – P. 10, line 8).

2. Add more recent published work: In the revised manuscript, we have added recent findings that report the roles of diverse environment factors affecting the formation of new particles (P3, line 10; P3, lines 17–24).

3. Typical lifetime of PM is generally longer than 2 days that used to identify the origin of measure air mass: We agree with this referee that the retention time of PM10 is known to be >2 days in the lower troposphere. The retention time for aerosol particles with a diameter <10 $\mu$m is known to be 3–5 days in the low troposphere (Mishra et al., 2004; Budhavant et al., 2015). Therefore, we have applied 3-, 4- and 5-day air mass back trajectories to identify the potential origin of MSA in the revised manuscript. When applying 3-, 4-, and 5-day air mass back trajectories, the number of samples that satisfy >90% retention in the Bellingshausen and Weddell Seas was less than 20% of the total MSA samples owing to its longer transport pathway. Inevitably, the air mass origin of MSA was divided into two sectors i.e., the Bellingshausen Sea sector (<58.8oW) and the Weddell Sea sector (>58.8oW) by selecting the air mass back trajectories with >50% retention in a given sector. We have added these explanations in the revised manuscript (P4, lines 21–24; P8, lines 20–25; P8, lines 30–33).

Minor scientific and technical issues

4. Need to modify some terminologies: We have changed "aerosols" to "aerosol particles" (P1, line 15) and "secondary organic aerosols" to "new particles" (P1, line 35) in

the revised manuscript.

5. The paper written by Kim et al. (2018) is a companion paper submitted to ACP and is under review (Kim et al., New particle formation events observed at the King Sejong Station, Antarctic Peninsula – Part 1: Physical characteristics and contribution to cloud condensation nuclei, Atmos. Chem. Phys. Discuss., https://doi.org/10.5194/acp-2018-1180, in review, 2018).

6. Why there are only 6 measurements points in the figure, although the measurements cover 8 year? Unfortunately, we could not obtain the CN data during the summer period in January and February 2014 owing to a malfunction of the analytical system. As we started observation in March 2009, the CN data for January and February 2009 were not included in the analysis. Therefore, six measurement points are represented in Fig. 5.

7. Clarify the meaning of DMSP-to-chlorophyll ratio: The conversion of cellular DMSP into DMS is controlled by not only the concentration of DMSP but also, more importantly, the DMSP cleavage enzyme. The phytoplankton species containing high cellular DMSP (i.e., high DMSP-to-chlorophyll ratio) mostly possess an enzyme that can convert cellular DMSP into DMS, whereas phytoplankton species containing low DMSP content (i.e., low DMSP-to-chlorophyll ratio) do not have a DMSP cleavage enzyme. Therefore, the DMSP-to-chlorophyll ratio is commonly used to explain the differences in taxonomic compositions affecting the oceanic DMS-production capacity (e.g., Belviso et al., 2000; Stefels et al., 2007; Tison et al., 2010; Park et al., 2014b and 2018). We have added more description to clarify the meaning of the DMSP-to-chlorophyll ratio in the revised manuscript (P4, line 34 – P5, line 4; P7, lines 15–40).

8. We have changed "DEB" to "DEC" (Figure S3 and S4 in the revised supplement).

Please also note the supplement to this comment:
https://www.atmos-chem-phys-discuss.net/acp-2018-1181/acp-2018-1181-AC2-

supplement.pdf

---

## Author Comment (AC3) · 10 Apr 2019

We thank Referee 3 for providing valuable comments on our manuscript. Our responses to this Referee's five scientific issues are provided below. The revised manuscript was uploaded in the form of a supplement

Scientific issues

1. Only 38% of the data are analyzed. Present a whole analysis, including air masses originating other sectors: We have a companion paper (Kim et al., New particle formation events observed at King Sejong Station, Antarctic Peninsula – Part 1: Physical

characteristics and contribution to cloud condensation nuclei) that describes the physical characteristics of the aerosol particles observed at the King Sejong station during the same period. The whole dataset of new particle formation events recorded for eight years was analyzed in detail in this companion paper. In short, Kim et al. (2018) evaluated the numerous physical parameters (e.g., occurrence, formation rate, growth rate, condensation sink, and source rate of condensable vapor) that originated from South America, Antarctic Peninsula, and the Bellingshausen and Weddell Seas, and determined that ∼80% of the new particle formation events were observed when the air mass originated from ocean domain (including the Bellingshausen and Weddell Seas) during the productive summer peiod. Therefore, we focused on the influence of marine biota on the formation of nanoparticles in this study. A total of 22,469 hourly mean number concentrations of nanoparticles were measured over the eight years. The linkage between new particle formation and environmental parameters is complicated, owing to the interplay among multiple sources and complicated processes. We have excluded the dataset that did not satisfy >90% retention over the two ocean domains to evaluate the relationship between new particle formation and oceanic biological characteristics. Thus, 8573 hourly mean number concentrations of nanoparticles were used for further analysis. To the best of our knowledge, it is based on the longest observation periods measured in the Antarctic site regarding this subject. We agree with this referee's comment that the previous version of this manuscript lacked explanation regarding the overall trend of nanoparticles, even though it was introduced in the companion paper. Therefore, we have added more description regarding "1) the general aspect of new particle formation events (e.g., frequency and potential origin) (P5, line 34 – P6, line 2)' 2) time-series transport history (Fig. 1c), 3) the reason why we focused on the dataset originating from two ocean domains (P1, lines 20–22; P6, lines 2–6) and 4) the limitations of the present study and the scope of future studies (P9, line 33 – P10, line 8)" in the revised manuscript.

2. As regards of the filter data, there are only few months of data presented, this should be stated in the abstract: A total of 84 MSA samples were analyzed daily during the

summer periods in 2013 and 2014. As this referee suggested, we have added this information to the Abstract (P1, line 29–30).

3. MSA concentration is too high (Issue 1) and local penguin emissions near the base should be discussed (Issue 2): (response to issue 1) We have added a new figure representing the daily MSA concentration and a brief explanation of MSA variation in the revised manuscript (Fig. 4a; P8, lines 16–27). The mean MSA concentration during the entire PM10 sampling period was 72.6 $\pm$ 99.1 ng m-3 (ranged from 4.2 to 657.0 ng m-3; 176.0 $\pm$ 186.2 (n=16), 68.1 $\pm$ 36.3 (n=25), 40.1 $\pm$ 30.9 (n=27), and 30.9 $\pm$ 26.2 (n=16) for Jan. 2013, Feb. 2013, Dec. 2013, and Jan. 2014, respectively), similar to the values observed at six Antarctic sites during the productive summer period. For example, the monthly mean MSA concentrations observed during the summer period were reported to be 59.3, 154.2, 63.0, 180, ~60, and ~100 ng m-3 in the Halley, Neumayer, Dumont d'Urville, Palmer, Zhongshan, and Marambio stations, respectively (e.g., Prospero et al., 1991; Minikin et al., 1998; Preunkert et al., 2007; Read et al., 2008; Zhang et al., 2015; Asmi et al., 2018). Note that most of the filter samples for MSA measurement were collected at a duration >3 days in these studies. As shown in Fig. 4a, an extremely high MSA concentration was observed between 14 Jan. 2013 and 17 Jan. 2013 (291.9, 386.6, 657.0, and 489.6 ng m-3 at a daily interval). During this period, the air masses originated from the Bellingshausen Sea (>90% air mass retention over the Bellingshausen Sea domain based on 2-, 3-, 4-, and 5-day air mass trajectories analysis). Such a high MSA concentration was also observed at the Antarctic site. For example, ~300 ng m-3 of MSA was reported at the Zhongshan Station (located in eastern Antarctic site) for the aerosol particles collected for a sampling duration of 10–15 days. Therefore, the high MSA concentration observed at the King Sejong Station appears to be due to the high biological productivities surrounding the observation site and the relatively shorter sampling duration during the study period. (response to issue 2) Over the study period of eight years, 16 days of new particle formation events out of 101 events were observed when the air mass originated from the Antarctic Peninsula. Penguin colonies are dispersed throughout the Antarctic Peninsula (Croxall et al.,

2002), and the emission of ammonia from these colonies could trigger the formation of nanoparticles (Weber et al., 1998; Croft et al., 2016). However, this issue was beyond the scope of the present study. In the revised manuscript, we have added more description to clarify the scope of the present study (P1, line 16; P1, lines 20–22; P5, line 34 – P6, line 6).

4. Need to add more recent findings regarding this issue: We have added more references that report the roles of diverse environment factors affecting the formation of new particles in the revised manuscript (P2, lines 17–24).

5. Need to carry on a more in-depth analysis with the whole data-set available: As we mentioned above, we have a companion paper that describes the physical characteristics of the aerosol particles observed at the King Sejong station during the entire sampling period. The number concentration of the nanoparticle was at its maximum during the productive summer period, and the frequency of new particle formation was highest when the air mass originated from the ocean domain. Therefore, we focused on the influence of marine biota on the formation of nanoparticles in this study. In the revised manuscript, we have added more description to clarify the scope of the present study (P1, line 16; P1, lines 20–22; P5, line 34 – P6, line 6). We have added more description and figures to support our key findings in the revised manuscript. In short, the explanation regarding the biological characteristic (e.g., DMSP-relevant process, feasibility of PHYSAT method and oceanic DMS production capacity) of the surrounding ocean was thoroughly complemented in the revised manuscript (P4, line 34 – P5, line 6; P7, lines 3–11; P7, line 16 – P8, line 40). Figures for 8-year CN records (CN2.5, CN10 and CN2.5-10), 8-year transport history at an hourly interval, sea-surface DMSP concentration, and daily MSA concentration have been newly added in the revised manuscript (Fig. 1, Fig. 3b, Fig. 4 and Fig. S4a). Furthermore, the limitations of the present study and the scope of future studies have also been added (P9, line 28 –P10, line 8).

Please also note the supplement to this comment:
https://www.atmos-chem-phys-discuss.net/acp-2018-1181/acp-2018-1181-AC3-supplement.pdf

[Figure]

**Supplement:**

[revised manuscript text omitted]

Several studies reported that DMSP and DMS were strongly linked to several environmental parameters such as solar radiation, sea-surface temperature, and mixing state of the sea surface (Vallina and Simo, 2007). *Gali* et al. (2015) developed a DMSP algorithm based on satellite-derived chlorophyll (to measure phytoplankton biomass) and the light exposure regime (to measure key environmental factors controlling DMSP production). In this algorithm, euphotic layer depth ($Z_{eu}$) and mixed layer depth (MLD) dataset were applied to establish a mixing state of the sea surface (stratified vs. mixed water column), and the variability in modeled and measured DMSP was improved by adding sea-surface temperature and $\log_{10}(Zeu/MLD)$ as predictors for the stratified and mixed subsets in the proposed algorithm. Additionally, a sub-model based on particulate inorganic carbon (PIC) was developed to complement DMSP diagnosis in coccolithophore blooms, where satellite chlorophyll concentration may not be reliable. The database was divided into three subsets including 'stratified water ($Z_{eu}/MLD > 1$)', 'mixed water ($Z_{eu}/MLD < 1$)' and 'undefined water ($Z_{eu}$ or MLD is unavailable)' based on the ratio between the euphotic layer depth ($Z_{eu}$) and the mixed layer depth (MLD). The $DMSP_t$ concentrations in stratified, mixed and undefined water were calculated using Equations (S1), (S2) and (S3), respectively:

$$\text{Log}_{10}DMSP_t = 1.70 + 1.14\log_{10}Chl_t + 0.44\log_{10}Chl_t^2 + 0.063SST - 0.0024SST^2 \tag{S1}$$
$$\text{Log}_{10}DMSP_t = 1.74 + 0.81\log_{10}Chl_t + 0.60\log_{10}(Z_{eu}/MLD) \tag{S2}$$
$$\log_{10}DMSP_t = -1.052 - 3.185\log_{10}PIC - 0.783(\log_{10}PIC)^2 \tag{S3}$$

The level-3 product of the Moderate Resolution Imaging Spectroradiometer on the Aqua (MODIS-Aqua) satellites was used for the chlorophyll concentration ($Chl_t$), sea surface temperature at nighttime (SST) and the calcite concentration (PIC). The monthly mixed layer depth (MLD) was retrieved by Monthly Isopycnal and Mixed-layer Ocean Climatology (MIMOC) at a resolution of 0.5°. All of the MODIS-Aqua products at a resolution of 4 km were averaged onto a 0.5° interval grid of MIMOC climatology to run the DMSPt algorithm. The euphotic layer depth ($Z_{eu}$) was calculated using satellite-derived chlorophyll data as shown in Equation (S4) (Morel et al., 2007).

$$\log_{10}Z_{eu} = 1.524 - 0.436\log_{10}Chl_t - 0.0145(\log_{10}Chl_t)^2 + 0.0186(\log_{10}Chl_t)^3 \tag{S4}$$

**Table S1.** Monthly average, 1 standard deviation and 95% confidence interval of nanoparticles (2.5–10 nm in diameter, $CN_{2.5-10}$) that originated from the Bellingshausen and Weddell Seas during the study period. A *t*-test was used to determine if there is a statistically significant difference between the means number concentration nanoparticles originated from the two selected ocean domains.

| | | Avg. | Std. | 95% confidence interval* | | *p*-value (*t*-test) |
| --- | --- | --- | --- | --- | --- | --- |
| | | | | Upper bound | Lower bound | |
| Jan. | Weddell sea | 332.0 | 920.7 | 196.3 | 782.4 | 0.0003 |
| | Bellingshausen sea | 835.9 | 2673.2 | 705.9 | 1004 | |
| Feb. | Weddell sea | 254.3 | 284.0 | 181.4 | 355.8 | 0.0010 |
| | Bellingshausen sea | 523.7 | 2130.7 | 417.2 | 695.4 | |
| Mar. | Weddell sea | 60.7 | 60.3 | 47.0 | 78.2 | < 0.0001 |
| | Bellingshausen sea | 166.6 | 550.3 | 142.6 | 208.5 | |
| Apr. | Weddell sea | 70.0 | 103.6 | 53.9 | 96.4 | 0.0245 |
| | Bellingshausen sea | 100.9 | 272.4 | 87.7 | 126.6 | |
| May | Weddell sea | 89.6 | 74.5 | 75.2 | 108.4 | < 0.0001 |
| | Bellingshausen sea | 45.2 | 56.2 | 41.0 | 50.9 | |
| Jun. | Weddell sea | 58.0 | 22.0 | NaN** | NaN | NaN |
| | Bellingshausen sea | 57.7 | 62.1 | 52.7 | 64.1 | |
| Jul. | Weddell sea | 22.9 | 17.9 | 15.1 | 35.1 | 0.0031 |
| | Bellingshausen sea | 42.8 | 56.8 | 36.8 | 51.6 | |
| Aug. | Weddell sea | - | - | NaN | NaN | NaN |
| | Bellingshausen sea | 58.3 | 78.6 | 47.5 | 73.6 | |
| Sep. | Weddell sea | 3.7 | - | NaN | NaN | NaN |
| | Bellingshausen sea | 97.9 | 85.9 | 91.0 | 105.5 | |
| Oct. | Weddell sea | 193.1 | 160.5 | NaN | NaN | NaN |
| | Bellingshausen sea | 129.0 | 405.1 | 110.3 | 197.7 | |
| Nov. | Weddell sea | 88.0 | 61.0 | 74.3 | 107.4 | < 0.0001 |
| | Bellingshausen sea | 176.7 | 331.9 | 154.0 | 214.3 | |
| Dec. | Weddell sea | 200.5 | 380.5 | 56.5 | 499.9 | 0.2111 |
| | Bellingshausen sea | 343.0 | 1138.8 | 277.6 | 449.0 | |

5    *confidence interval was estimated by bootstrap method that was calculated from 10,000 subsamples generated by random sampling with replacement from monthly $CN_{2.5-10}$ data.
**number of monthly $CN_{2.5-10}$ data <10 was excluded from the bootstrap and *t*-test.

[Figure]

(a)     (b)

Sea ice    Land    Ocean

**Figure S1:** Percentage of the hourly trajectory points that passed over the three major areas surrounding the observation site including sea-ice (red), land (yellow) and ocean (blue) to the total number of hourly trajectory points in the 2-day air-mass trajectory during (a) the overall period (from January to December) and (b) the austral summer period (December, January and February) between March 2009 and November 2016.

[Figure]

**Figure S2:** (a) Monthly mean chlorophyll concentration around the observation site between 2009 and 2016 (55°S–65°S, 40°W–80°W). (b) Monthly mean chlorophyll concentration for the two selected ocean domains including the Weddell (red symbols; 55°S–65°S, 40°W–60°W) and Bellingshausen (blue symbols; 55°S–65°S, 60°W–80°W) seas during the phytoplankton bloom period (October–February). Note that the monthly mean chlorophyll concentration was not available from May to August due to insufficient satellite-derived values (less than 10%) during the austral winter period.

[Figure]

**Figure S3:** The percentage of the dominant phytoplankton groups in the two ocean domains including (a) the Bellingshausen and (b) Weddell seas estimated using the PHYSAT method with the climatology map obtained from SeaWiFS archive during the austral summer period.

[Figure]

**Figure S4:** (a) Monthly mean DMSP concentration and (b) monthly mean DMSP-to-chlorophyll ratio in the Bellingshausen (blue bars) and Weddell (red bars) Seas during the austral summer period between March 2009 and November 2016.

[Figure]

**Figure S5:** Mixed layer depth retrieved using the Monthly Isopycnal and Mixed-layer Ocean Climatology (MIMOC) during the austral summer period surrounding King Sejong Station (red star symbol).

---

## Author Comment (AC4) · 10 Apr 2019

We thank to Dr. Allan for providing insightful suggestions that improved the readability of our revised manuscript. Our responses to Dr. Allan's comments are stated below. The revised manuscript was uploaded in the form of a supplement

Scientific comment

1. Need to add explanation on the roles of iodine compounds in new particle formation events: We agree that iodine-containing compounds which produced by both biotic and abiotic processes are one of the key compounds contributing to new particle

formation in the Arctic and Antarctic regions. Unfortunately, we do not have observational evidence (i.e., chemical analysis of halogen compounds) to support the roles of iodine-compounds in formation of new particles in this study. In the revised manuscript, we have added recent findings regarding the potential roles of halogen compounds in the 'Introduction' part (P2, lines 17–22). Molecular-scale measurements of chemical species (e.g., sulfur-, nitrogen-, and halogen-containing compounds) involved in nucleation processes are required to provide direct evidence for the contribution of these compounds to the formation and growth of aerosol particles and to understand their climate feedback roles in the remote marine environment. Therefore, we have added a short paragraph indicating the limitations of the present study and the scope of future studies in this regard (P9, line 33 – P10, line 8).

Please also note the supplement to this comment:
https://www.atmos-chem-phys-discuss.net/acp-2018-1181/acp-2018-1181-AC4-supplement.pdf

―――――――――――――――――――

**Supplement:**

[revised manuscript text omitted]

Several studies reported that DMSP and DMS were strongly linked to several environmental parameters such as solar radiation, sea-surface temperature, and mixing state of the sea surface (Vallina and Simo, 2007). *Gali* et al. (2015) developed a DMSP algorithm based on satellite-derived chlorophyll (to measure phytoplankton biomass) and the light exposure regime (to measure key environmental factors controlling DMSP production). In this algorithm, euphotic layer depth ($Z_{eu}$) and mixed layer depth (MLD) dataset were applied to establish a mixing state of the sea surface (stratified vs. mixed water column), and the variability in modeled and measured DMSP was improved by adding sea-surface temperature and $\log_{10}(Zeu/MLD)$ as predictors for the stratified and mixed subsets in the proposed algorithm. Additionally, a sub-model based on particulate inorganic carbon (PIC) was developed to complement DMSP diagnosis in coccolithophore blooms, where satellite chlorophyll concentration may not be reliable. The database was divided into three subsets including 'stratified water ($Z_{eu}/MLD > 1$)', 'mixed water ($Z_{eu}/MLD < 1$)' and 'undefined water ($Z_{eu}$ or MLD is unavailable)' based on the ratio between the euphotic layer depth ($Z_{eu}$) and the mixed layer depth (MLD). The $DMSP_t$ concentrations in stratified, mixed and undefined water were calculated using Equations (S1), (S2) and (S3), respectively:

$$\text{Log}_{10}DMSP_t = 1.70 + 1.14\log_{10}Chl_t + 0.44\log_{10}Chl_t^2 + 0.063SST - 0.0024SST^2 \tag{S1}$$
$$\text{Log}_{10}DMSP_t = 1.74 + 0.81\log_{10}Chl_t + 0.60\log_{10}(Z_{eu}/MLD) \tag{S2}$$
$$\log_{10}DMSP_t = -1.052 - 3.185\log_{10}PIC - 0.783(\log_{10}PIC)^2 \tag{S3}$$

The level-3 product of the Moderate Resolution Imaging Spectroradiometer on the Aqua (MODIS-Aqua) satellites was used for the chlorophyll concentration ($Chl_t$), sea surface temperature at nighttime (SST) and the calcite concentration (PIC). The monthly mixed layer depth (MLD) was retrieved by Monthly Isopycnal and Mixed-layer Ocean Climatology (MIMOC) at a resolution of 0.5°. All of the MODIS-Aqua products at a resolution of 4 km were averaged onto a 0.5° interval grid of MIMOC climatology to run the DMSPt algorithm. The euphotic layer depth ($Z_{eu}$) was calculated using satellite-derived chlorophyll data as shown in Equation (S4) (Morel et al., 2007).

$$\log_{10}Z_{eu} = 1.524 - 0.436\log_{10}Chl_t - 0.0145(\log_{10}Chl_t)^2 + 0.0186(\log_{10}Chl_t)^3 \tag{S4}$$

**Table S1.** Monthly average, 1 standard deviation and 95% confidence interval of nanoparticles (2.5–10 nm in diameter, $CN_{2.5-10}$) that originated from the Bellingshausen and Weddell Seas during the study period. A *t*-test was used to determine if there is a statistically significant difference between the means number concentration nanoparticles originated from the two selected ocean domains.

| | | Avg. | Std. | 95% confidence interval* | | *p*-value (*t*-test) |
| --- | --- | --- | --- | --- | --- | --- |
| | | | | Upper bound | Lower bound | |
| Jan. | Weddell sea | 332.0 | 920.7 | 196.3 | 782.4 | 0.0003 |
| | Bellingshausen sea | 835.9 | 2673.2 | 705.9 | 1004 | |
| Feb. | Weddell sea | 254.3 | 284.0 | 181.4 | 355.8 | 0.0010 |
| | Bellingshausen sea | 523.7 | 2130.7 | 417.2 | 695.4 | |
| Mar. | Weddell sea | 60.7 | 60.3 | 47.0 | 78.2 | < 0.0001 |
| | Bellingshausen sea | 166.6 | 550.3 | 142.6 | 208.5 | |
| Apr. | Weddell sea | 70.0 | 103.6 | 53.9 | 96.4 | 0.0245 |
| | Bellingshausen sea | 100.9 | 272.4 | 87.7 | 126.6 | |
| May | Weddell sea | 89.6 | 74.5 | 75.2 | 108.4 | < 0.0001 |
| | Bellingshausen sea | 45.2 | 56.2 | 41.0 | 50.9 | |
| Jun. | Weddell sea | 58.0 | 22.0 | NaN** | NaN | NaN |
| | Bellingshausen sea | 57.7 | 62.1 | 52.7 | 64.1 | |
| Jul. | Weddell sea | 22.9 | 17.9 | 15.1 | 35.1 | 0.0031 |
| | Bellingshausen sea | 42.8 | 56.8 | 36.8 | 51.6 | |
| Aug. | Weddell sea | - | - | NaN | NaN | NaN |
| | Bellingshausen sea | 58.3 | 78.6 | 47.5 | 73.6 | |
| Sep. | Weddell sea | 3.7 | - | NaN | NaN | NaN |
| | Bellingshausen sea | 97.9 | 85.9 | 91.0 | 105.5 | |
| Oct. | Weddell sea | 193.1 | 160.5 | NaN | NaN | NaN |
| | Bellingshausen sea | 129.0 | 405.1 | 110.3 | 197.7 | |
| Nov. | Weddell sea | 88.0 | 61.0 | 74.3 | 107.4 | < 0.0001 |
| | Bellingshausen sea | 176.7 | 331.9 | 154.0 | 214.3 | |
| Dec. | Weddell sea | 200.5 | 380.5 | 56.5 | 499.9 | 0.2111 |
| | Bellingshausen sea | 343.0 | 1138.8 | 277.6 | 449.0 | |

5    *confidence interval was estimated by bootstrap method that was calculated from 10,000 subsamples generated by random sampling with replacement from monthly $CN_{2.5-10}$ data.
**number of monthly $CN_{2.5-10}$ data <10 was excluded from the bootstrap and *t*-test.

[Figure]

(a)     (b)

Sea ice    Land    Ocean

**Figure S1:** Percentage of the hourly trajectory points that passed over the three major areas surrounding the observation site including sea-ice (red), land (yellow) and ocean (blue) to the total number of hourly trajectory points in the 2-day air-mass trajectory during (a) the overall period (from January to December) and (b) the austral summer period (December, January and February) between March 2009 and November 2016.

[Figure]

**Figure S2:** (a) Monthly mean chlorophyll concentration around the observation site between 2009 and 2016 (55°S–65°S, 40°W–80°W). (b) Monthly mean chlorophyll concentration for the two selected ocean domains including the Weddell (red symbols; 55°S–65°S, 40°W–60°W) and Bellingshausen (blue symbols; 55°S–65°S, 60°W–80°W) seas during the phytoplankton bloom period (October–February). Note that the monthly mean chlorophyll concentration was not available from May to August due to insufficient satellite-derived values (less than 10%) during the austral winter period.

[Figure]

**Figure S3:** The percentage of the dominant phytoplankton groups in the two ocean domains including (a) the Bellingshausen and (b) Weddell seas estimated using the PHYSAT method with the climatology map obtained from SeaWiFS archive during the austral summer period.

[Figure]

**Figure S4:** (a) Monthly mean DMSP concentration and (b) monthly mean DMSP-to-chlorophyll ratio in the Bellingshausen (blue bars) and Weddell (red bars) Seas during the austral summer period between March 2009 and November 2016.

[Figure]

**Figure S5:** Mixed layer depth retrieved using the Monthly Isopycnal and Mixed-layer Ocean Climatology (MIMOC) during the austral summer period surrounding King Sejong Station (red star symbol).

---

## Author Comment (AC5) · 10 Apr 2019

The authors are very grateful to the careful review and valuable recommendations from three referees and Dr. Allan which have considerably improved the readability of the revised manuscript. Here we attach the revised manuscript with tracked changes.

Please also note the supplement to this comment:
https://www.atmos-chem-phys-discuss.net/acp-2018-1181/acp-2018-1181-AC5-supplement.pdf

2018.

**Supplement:**

**New particle formation events observed at the King Sejong Station, Antarctic Peninsula – Part 2: Link with the oceanic biological activities**

5     Eunho Jang[1, 2,*], Ki-Tae Park[1,*], Young Jun Yoon[1], Tae-Wook Kim[3], Sang-Bum Hong[1], Silvia Becagli[4], Rita Traversi[4], Jaeseok Kim[5], Yeontae Gim[1]

[1]Korea Polar Research Institute, 26 Songdomirae-ro, Yeonsu-gu, Incheon 21990, South Korea
[2]University of Science and Technology, 217 Gajeong-ro, Yuseong-gu, Daejeon 34113, South Korea
[3]Division of Environmental Science and Ecological Engineering, Korea University, Seoul, South Korea
10     [4]Department of Chemistry "Ugo Schiff", University of Florence, via della Lastruccia, 3, Sesto F.no (FI), 50019, Italy.
[5]Korea Research Institute of Standards and Science, 267 Gajeong-ro, Yuseong-gu, Daejeon 34113, South Korea
[*]These authors contributed equally to this work

*Correspondence to*: Ki-Tae Park (ktpark@kopri.re.kr)

15     **Abstract.** Marine biota is an important source of atmospheric  aerosol particles in the remote marine atmosphere. However, the relationship between new particle formation and marine biota is poorly quantified. Long-term observations (from 2009 to 2016) of the physical properties of atmospheric aerosol particles measured at the Antarctic Peninsula (King Sejong Station; 62.2°S, 58.8°W) and satellite-derived estimates of the biological characteristics were analyzed to identify the link between new particle formation and marine biota. New particle

20     formation events in the Antarctic atmosphere showed distinct seasonal variations, with the highest values  occurred when the air mass originated from the ocean domain during productive austral summer (December, January and February). Interestingly, new particle formation events were more frequent in the air masses that originated from the Bellingshausen Sea than in those that originated from the Weddell Sea. The monthly mean number concentration of nanoparticles (2.5–10 nm in diameter) was >2-fold when the air masses

25     passed over the Bellingshausen Sea than the Weddell Sea, whereas the biomass of phytoplankton in the Weddell Sea was more than ~70% higher than that of the Bellingshausen Sea during the austral summer period. Dimethyl sulfide (DMS) is of marine origin and its oxidative products are known to be one of the major components in the formation of new particles. Both satellite-derived estimates of the biological characteristics (dimethylsulfoniopropionate (DMSP; precursor of DMS) and phytoplankton taxonomic composition) and in situ

30     methanesulfonic acid ( 84 daily measurements during the summer period in 2013 and 2014) analysis revealed that DMS(P)-rich phytoplankton were more dominant in the Bellingshausen Sea than in the Weddell Sea. Furthermore, the number concentration of nanoparticles was positively correlated with the biomass of phytoplankton during the period when DMS(P)-rich phytoplankton predominate. These results indicate that oceanic DMS emissions could play a key role in the formation of new particles; moreover, the

35     taxonomic composition of phytoplankton could affect the formation of  new particles in the Antarctic Ocean.

**메모 포함[KTP1]:** (response to referee #2)
Q4. As suggested, we have changed "*aerosols*" to "*aerosol particles*".

**메모 포함[WU2]:** (response to referee #1 and #3)
We have added this sentence to clarify the scope of this study.

**메모 포함[KTP3]:** (response to referee #1 and #3)
We have changed "*occurring during austral summer*" to "*occurred when the air mass originated from the ocean domain during the productive austral summer*" to clarify the scope of this study.

**메모 포함[WU4]:** (response to referee #3)
**Q2. As regards of the filter data, there are only few months of data presented, this should be stated in the abstract:** As this referee suggested, we have added this information to the Abstract.

**메모 포함[KTP5]:** (response to referee #2)
Q4. As suggested, we have changed "*secondary organic aerosols*" to "*new particles*".

**1 Introduction**

[revised manuscript text omitted]

메모 포함[KTP8]: (response to referee #3)
**Q1. Only 38% of the data are analyzed. Present a whole analysis, including air masses originating other sectors:** We have added new figure that represent the 8-year nanoparticle measurement and corresponding air mass transport history.

메모 포함[KTP9]: (response to referee #2)
**Q5.** This paper is a companion paper submitted to ACP and is under review (https://doi.org/10.5194/acp-2018-1180).

2014). Half of a 47-mm Teflon filter was used to measure major ions including MSA. The MSA (and the other ions) collected on the filter was extracted into about 10 mL (18 MΩ Milli-Q) in ultrasonic bath for 20 min. MSA was determined by an ion chromatography system (Dionex, Thermo Fisher Scientific Inc.) following the procedure described by Becagli et al. (2012). For MSA, reproducibility on real samples was better than 5%. Filter

5    blank concentrations for methanesulphonate were always below the detection limit.

**2.2. Air-mass back trajectories**

The air mass back trajectories and meteorological parameters were obtained using the Hybrid Single-Particle Lagrangian Integrated Trajectories model (Draxler and Hess, 1998). In general, the growth rate of sub-micron

10   particles in the remote marine environment ranges from 0.2 to 5.0 nm h$^{-1}$ (Järvinen et al., 2013; Weller et al., 2015; Kerminen et al., 2018), and the mean growth rate of the aerosol particles measured at the King Sejong Station was $0.68 \pm 0.27$ nm h$^{-1}$ during the eight years (Kim et al., 2018). Therefore, the 2-day air mass back trajectories and hourly positions were determined and combined with satellite-derived geographical information to identify the travel history of the air mass arriving at the observation site. Daily geographical information on sea-ice, land, and

15   ocean area was obtained from the Sea Ice Index at a 25-km resolution provided by the National Snow and Ice Data Center (NSIDC). The oceanic region adjacent to the observation site was surrounded by two different ocean basins, namely, the Bellingshausen and Weddell seas. To evaluate the influence of the oceanic biological characteristics on the occurrence of new particle formation, we limited our analysis to the air masses that had exposure predominantly to the ocean area. Specifically, the origin of the hourly air mass arriving at the observation

20   site was divided into two ocean domains (i.e., the Bellingshausen and Weddell seas). Then, all air mass back trajectories were grouped into one of the two ocean domains by only selecting the 2-day air mass back trajectories that had >90% retention in a given ocean domain. A total of 84 PM$_{10}$ samples for MSA analysis were collected daily during the summer periods in 2013 and 2014. The retention time of the aerosol particles with a diameter <10 μm is known to be approximately 3–5 days in the

25   atmosphere (Mishra et al., 2004; Budhavant et al., 2015). Therefore, 3-, 4-, and 5-day air mass back trajectories were applied to identify the potential origin of MSA during the sampling period.

**2.3. Phytoplankton biomass, DMSP and taxonomic composition analysis**

Satellite-derived ocean color provide a good measure of analyzing the phytoplankton characteristics of the Southern Ocean (Siegel et al., 2013; Haentjens et al., 2017). The phytoplankton biomass of the two ocean domains

30   was estimated by calculating the chlorophyll concentration from the Moderate Resolution Imaging Spectroradiometer on the Aqua (MODIS-Aqua) satellites at 4 km resolution during the study period (2009–2016). The trajectory concentration of the air masses originating from the two ocean domains was calculated from the ratio of the number of hourly trajectory points passing over each grid cell (1° × 1°) to the total number of hourly trajectory points (Kim et al., 2011), as shown in Fig. 1a 2a. We limited our analysis of satellite-derived chlorophyll

35   concentration to the ocean area for which the trajectory concentration was approximately over 0.1% (55–65°S, 40–60°W for the Weddell Sea and 55–65°S, 60–80°W for the Bellingshausen Sea). DMSP is produced by marine phytoplankton and is the most important precursor of oceanic DMS production. However, the dependence of the

메모 포함[KTP10]: (response to referee #1)
**Q6.** As suggested, we have replaced "*below detection*" to "*below detection limit*".

메모 포함[WU11]: (response to referee #2)
**Q3. Typical lifetime of PM is generally longer than 2 days that used to identify the origin of measure air mass:** We agree with referee #2 that retention time of PM$_{10}$ is known to be >2 days in the lower troposphere. Therefore, we have applied 3-, 4- and 5-day air mass back trajectories to identify the potential origin of MSA in the revised manuscript.

oceanic DMS emission on phytoplankton biomass and DMSP concentration is not straightforward owing to the strong variabilities across taxonomic groups and its interplay with environmental factors. Nevertheless, temporal and spatial distribution of sea-surface DMSP could be an indicator of contemporary DMS emission. In particular, the DMSP-to-chlorophyll ratio could represent the potential DMS production capacity of the ocean because the phytoplankton species with higher cellular DMSP content (i.e., higher DMSP-to-chlorophyll ratio) mostly possess an enzyme that can convert cellular DMSP into DMS, whereas phytoplankton species containing lower DMSP content (i.e., lower DMSP-to-chlorophyll ratio) do not have a DMSP cleavage enzyme (Stefels et al., 2007, Park et al., 2014b and 2018). The total DMSP concentration in the sea-surface was estimated using the algorithm developed by Galí et al. (2015). The algorithm for the total DMSP concentration was based on the satellite-derived chlorophyll concentration and the light exposure regime (see Supplement for more information). We estimated the taxonomic phytoplankton composition of the two ocean domains using the PHYSAT method. This method is a bio-optic model that was specifically developed to identify the dominant phytoplankton groups from ocean color measurements. Phytoplankton groups are generally characterized by specific pigment, shape and size that have different light scattering and absorption properties (Alvain et al., 2005). Therefore, the PHYSAT method can be used to classify sea-surface phytoplankton into five groups including diatoms, Prochlorococcus, nanoeucaryotes, Synechococcus and phaeocystis simultaneously with the chlorophyll concentration (Alvain et al., 2008; Mustapha et al., 2014). The monthly data set of phytoplankton groups at a resolution of 9 km was obtained from the PHYSAT database (http://log.univ-littoral.fr/Physat), and was estimated between 1997 and 2010. The period of the PHYSAT analysis did not coincide with the period of aerosol analysis. This is because the PHYSAT method was developed and calibrated based on daily global data derived from Sea-viewing Wide Field-of-view Sensor (SeaWiFS) that had been operated from September 1997 to December 2010. The PHYSAT method was first developed in 2005 and was used to classify sea-surface phytoplankton into four groups: diatoms, Prochlorococcus, nanoeucaryotes, and Synechococcus (Alvain et al., 2005). Subsequently, the modified PHYSAT method, which can estimate the contribution of the phaeocystis group, was reported in 2008 (Alvain et al., 2008). The PHYSAT method was developed and calibrated based on global data obtained from the sea-viewing wide field-of-view sensor (SeaWiFS) operated from September 1997 to December 2010. In this study, the monthly dataset of five phytoplankton groups at a resolution of 9 km was obtained from the PHYSAT database (http://log.univ-littoral.fr/Physat) estimated using climatology over the SeaWiFS period (1997–2010). Note that "dominant" has been defined as situations in which a given phytoplankton group is a major contributor to the total pigment in a given 9 km resolution (Alvain et al., 2005 and 2008).

**3 Results and Discussion**

**3.1. Seasonal variabilities of nanoparticles at King Sejong Station**

The number concentration of aerosol particles increased gradually from early spring, peaked in the austral summer period (December, January and February) and then began to decrease (Fig. S1 1a). The number concentration of nanoparticles (2.5–10 nm in diameter, $CN_{2.5-10}$), which is an indication of newly formed particles, also shows distinct seasonal variation (Fig. 1b). The physical aspects of new particle formation event events (e.g., number of new particle formation event days, formation, and growth rate) observed at the same site are explained in detail

메모 포함[KTP12]: (response to referee #1 and #2)
**Q1 and Q7: Clarify the meaning of DMSP to chlorophyll ratio:** We have added brief explanation regarding DMSP and its biological aspects. More detailed description was added to clarify the meaning of the DMSP to chlorophyll ratio in chapter 3.2.

메모 포함[KTP13]: (response to referee #1)
**Q1. How representative is the PHYSAT database for the period 1997-2010 with regard to your observation period?** The PHSYAT calculated by SeaWiFS climatology map is the most widely used algorithm for the estimation of taxonomic composition of marine phytoplankton. We have revised and added the explanation regarding the PHYSAT method in chapter 2.3 and 3.2.

메모 포함[KTP14]: (response to referee #1)
**Q3. Provide clear description regarding the results for the PHYSAT analysis:** We have added the meaning of "*dominant*" used in the PHYSAT method.

in Kim et al. (2018) . The observation site is surrounded by ocean, sea-ice, and land domains, which may influence new particle formation in different ways. The 2-day air mass back trajectory combined with geographical information showed that approximately

5  66% of the hourly trajectory points were assigned to the ocean, followed by sea-ice (29%) and land (6%) during the entire study period (Fig. 1c and Fig. S1a). The percentage of hourly trajectory points that passed over the ocean domain were at their maximum during the summer period (79%) when the extent of sea-ice was at its minimum (Fig. S1b). Kim et al. (2018) reported that a total of 101 days were defined as new particle formation events during the eight years and 80 days of new particle formation events occurred when the air mass originated

10  from the ocean domain. Furthermore, 16 days of new particle formation events were observed for the air masses originating from the Antarctic Peninsula. The remaining five days of events were considered as those of South American origin (three events) and undefined (two events) (see Kim et al. 2018 for the detailed definition and categorization of new particle formation events). The relationship between new particle formation and environmental parameters is complicated, owing to the interplay among multiple sources and complicated

15  processes. The number concentration of the nanoparticles was at its maximum during the productive summer period, and the frequency of new particle formation was the highest when the air mass originated from the ocean domain. Therefore, we focused on the influence of marine biota on the formation of nanoparticles. The hourly mean concentration of nanoparticles matched with the hourly air mass back trajectory in this study. A total of 22,469 hourly mean number concentrations of nanoparticles were measured above the Antarctic atmosphere over

20  the eight years. Approximately 38.2% of the hourly mean number concentration of nanoparticles, which satisfy the >90% retention of hourly trajectory points over the two ocean domains, were used to estimate the link between new particle formation and the oceanic biological characteristics around the observation site. The remaining 61.8% of the hourly mean number concentration of nanoparticles, which do not satisfy the >90% retention over the two ocean domains, were excluded from further analysis. Interestingly, the monthly mean number concentration of

25  nanoparticles that originated from the Bellingshausen Sea was highest in January (835.9 ± 2673.2 cm$^{-3}$) and ~2.5-times greater than that which originated from the Weddell Sea (332.0 ± 920.7 cm$^{-3}$; Fig.  2b and Table S1). The differences in the number concentration of nanoparticles that originated from the two ocean domains were particularly noticeable during the austral summer period (567.5 ± 249.3 cm$^{-3}$ for the Bellingshausen Sea and 262.3 ± 66.1 cm$^{-3}$ for the Weddell Sea). However, the differences were not evident between March and November (97.2

30  ± 51.1 cm$^{-3}$ for the Bellingshausen Sea and 73.2 ± 56.9 cm$^{-3}$ for the Weddell Sea; Fig.  2b).

**3.2. Biological characteristics surrounding the observation site**

In general, the abundance and composition of phytoplankton show distinct spatial and seasonal variation in the Antarctic Ocean (Sullivan et al., 1993). Primary production in the Antarctic Ocean is strongly controlled by various factors such as iron limitation, light availability and mixed layer depth (Arrigo et al., 1999; Sedwick et al.,

35  2007; Park et al., 2013a). The composition of the phytoplankton community is poorly studied in the Antarctic Ocean except for the marginal zone at the Antarctic Peninsula. Nevertheless, both phaeocystis and diatoms (mainly *Phaeocystis antarctica* and *Fragilariopsis cylindrus*) are well known as dominant phytoplankton groups in the Antarctic Ocean during the phytoplankton bloom period (Kropuenske et al., 2009; Arrigo et al., 2010). Both

메모 포함[KTP15]: (response to referee #3)
**Q1 and Q5. Present a whole analysis, including air masses originating other sectors:** We have a companion paper (Kim et al., New particle formation events observed at King Sejong Station, Antarctic Peninsula – Part 1: Physical characteristics and contribution to cloud condensation nuclei) that describes the physical characteristics of the aerosol particles observed at the King Sejong station during the same period. The whole dataset of new particle formation events recorded for eight years was analyzed in detail in this companion paper. We agree with this referee's comment that the previous version of this manuscript lacked explanation regarding the overall trend of nanoparticles, even though it was introduced in the companion paper. Therefore, we have added more description regarding "1) the general aspect of new particle formation events (e.g., frequency and potential origin)' 2) time-series transport history (see Fig. 1c), and 3) the reason why we focused on the dataset originating from two ocean domains" in the revised manuscript

메모 포함[WU16]: (response to referee #1)
**Q4. Did the remaining 61.8% of the data indicate any significant link to the origin of the air masses?** The remaining 61.8% of hourly mean number concentration of nanoparticles may have undergone more complicated transport history. Therefore, it was not easy to find a strong relationship between the formation of nanoparticles and the potential source origin (e.g., marine biota, sea-ice extent, and penguin colony). In the revised manuscript, we have added more description to clarify the scope of the present study in chapter 3. Furthermore, the limitation of this study has been added in chapter 4.

[revised manuscript text omitted]

메모 포함[KTP17]: (response to referee #1)
**Q1. The discussion presented in Chapters 3 is not conclusive:** We have added this paragraph to improve the readability.

메모 포함[KTP18]: (response to referee #1)
**Q1. How representative is the PHYSAT database for the period 1997-2010 with regard to your observation period?** We have replaced "*which was applied from 1997 to 2010*" to "*which was estimated by SeaWiFS climatology map*" to clarify the meaning.

메모 포함[KTP19]: (response to referee #1)
**Q3. Provide clear description regarding the results for the PHYSAT analysis:** We have replaced "*~35% of the satellite pixels were dominated by diatoms*" to "*the dominance of the diatom was ~35%.... and Synechococcus (14%)*".

메모 포함[KTP20]: (response to referee #1)
**Q1. How representative is the PHYSAT database for the period 1997-2010 with regard to your observation period?** An additional paragraph explaining the feasibility of the SeaWiFS-based PHYSAT method has been added.

메모 포함[WU21]: (response to referee #1)
**Q1. The description regarding DMSP-to-chlorophyll ratio is confusing:** We have added a paragraph explaining the DMSP-relevant processes and their association with marine biota to clarify the meaning of the DMSP-to-chlorophyll ratio in this chapter.

cleavage (Simó, 2001; Stefels et al., 2007). However, a larger proportion of dissolved DMSP is assimilated into bacterial tissues through demethylation processes, which do not produce gaseous DMS (Todd et al., 2007; Reisch et al., 2011). A direct correlation between the local chlorophyll concentration and atmospheric DMS mixing ratio in the absence of lag periods was observed in the Arctic Ocean where *Phaeocystis pouchetii* (containing both high cellular DMSP and DMSP cleavage enzyme) dominates (Park et al., 2013b). Moreover, the DMS production capacity in the Arctic Ocean was more significantly controlled by the abundance of DMSP-rich phytoplankton than the total biomass of phytoplankton (Park et al., 2018). These results indicate that the blooming of phytoplankton species containing higher cellular DMSP content results in a much higher DMS production capacity than the blooming of DMSP-poor phytoplankton species. Therefore, the DMSP-to-chlorophyll ratio is commonly used to explain the differences in taxonomic compositions affecting the oceanic DMS-production capacity (e.g., Belviso et al., 2000; Stefels et al., 2007; Tison et al., 2010; Park et al., 2014b and 2018). In particular, *Phaeocystis antarctica* was reported to be a dominant species in terms of DMS production in the Antarctic Ocean during the bloom period (Gibson et al., 1990; Schoemann et al., 2005), exhibiting a cellular DMSP concentration in phaeocystis several times that of diatoms (Hatton and Wilson, 2007; Stefels et al., 2007). The sea-surface DMSP concentration surrounding the observation site was estimated using a newly developed algorithm and was 30% higher in the Weddell Sea than in the Bellingshausen Sea during the summer period, possibly owing to intense blooming of DMSP-containing diatoms in the Weddell Sea (Fig. 3b and Fig S4a). This could illustrate that, despite having lower cellular DMSP content than phaeocystis, diatoms dominated the overall DMSP production in the Weddell Sea owing to their much larger biomass. However, the DMSP to chlorophyll ratio in the Bellingshausen Sea ($110.2 \pm 27.8$ mmol g$^{-1}$) was ~2-fold higher than that of the Weddell Sea ($72.2 \pm 8.3$ mmol g$^{-1}$) between December and February in 2009–2016 (Fig. 2c 3d and Fig. S5 S4b), possibly owing to the relatively higher contribution of the DMSP-rich phaeocystis group in the Bellingshausen Sea.

**3.3. Influence of phytoplankton on aerosol formation**

Biogenic trace gases produced by marine phytoplankton (i.e., DMS, isoprene, and halogenated gases) are known to be the key compounds contributing to the formation of new particles in the remote marine environment; however, quantifying the relationship between new particle formation events and marine biology is a major challenge (Brooks and Thornton, 2018). MSA in the marine atmosphere forms exclusively from the photooxidation of DMS, and shows strong seasonal variation (Ayers and Gras, 1991; Savoie et al., 1993; Preunkert et al., 2008). A previous study has reported that the highest values for both MSA and the scattering Ångström exponent (SAE; qualitative examination of the aerosol optical mean size) were observed at the Marambio Station (64.3°S, 56.6°W) on the Antarctic Peninsula during austral summer in 2013–2015 (Asmi et al., 2018). The MSA concentration of the fine aerosol particles measured at the King Sejong Station during the summer period in 2013 and 2014 was broadly consistent with the number concentration of nanoparticles. The MSA concentration shows distinct daily variations. The mean MSA concentration was $72.6 \pm 99.1$ ng m$^{-3}$ (ranged from 4.2 to 657.0 ng m$^{-3}$) (Fig. 4a), similar to the values observed at six Antarctic sites during the productive summer period (e.g., Prospero et al., 1991; Minikin et al., 1998; Preunkert et al., 2007; Read et al., 2008; Zhang et al., 2015; Asmi et al., 2018). To identify the potential origin of MSA, air mass back trajectories were determined and the retention time above

메모 포함[KTP22]: (response to referee #1)
**Q1. The description regarding DMSP-to-chlorophyll ratio is confusing. Q2. Need to compare the absolute DMSP concentration, not just the DMSP-to-chlorophyll ratio:** We have added more detailed description regarding the DMSP-relevant processes and their association with marine biota to clarify the meaning of DMSP to chlorophyll ratio. As this referee suggested, we have added figures for sea-surface DMSP concentration near the observation site (Fig. 3b and Fig. S4a). A brief explanation regarding the DMSP concentration has also been added.

메모 포함[KTP23]: (response to referee #1)
**Q1. The discussion presented in Chapters 3 is not conclusive:** We have added this paragraph to improve the readability.

메모 포함[KTP24]: (response to referee #1)
**Q1. The discussion presented in Chapters 3 is not conclusive:** We have added this paragraph to improve the readability.

메모 포함[KTP25]: (response to referee #1)
**Q5. Provide daily MSA concentration data:** We have added daily MSA concentration data in the revised manuscript (Fig. 4a). A brief explanation of MSA variation has also been added.

(response to referee #3)
**Q3. MSA concentration is too high:** The mean MSA concentration ($72.6 \pm 99.1$ ng m$^{-3}$) during the entire sampling period was similar to the values observed at six Antarctic sites during the productive summer period. For example, the monthly mean MSA concentrations observed during the summer period were reported to be 59.3, 154.2, 63.0, 180, ~60, and ~100 ng m$^{-3}$ in the Halley, Neumayer, Dumont d'Urville, Palmer, Zhongshan, and Marambio stations, respectively (e.g., Prospero et al., 1991; Minikin et al., 1998; Preunkert et al., 2007; Read et al., 2008; Zhang et al., 2015; Asmi et al., 2018).

each domain was averaged for the corresponding 24 h sampling time. When applying 3-, 4-, and 5-day air mass back trajectories, the number of samples that satisfy >90% retention in the Bellingshausen and Weddell Seas was less than 20% of the total MSA samples owing to its longer transport pathway. Inevitably, the air mass origin of MSA was divided into two sectors i.e., the Bellingshausen Sea sector (<58.8ºW) and the Weddell Sea sector (>58.8ºW) by selecting the air mass back trajectories with >50% retention in a given sector. Although the limited number of samples of MSA (84 samples at daily intervals) collected during the summer periods in 2013 and 2014 may not be sufficient to identify its source origin exactly, the MSA concentration also showed distinct differences depending on the air mass origin. The inflow of the air masses from the Bellingshausen Sea increased the concentration of MSA in the aerosol particles. Notably, the MSA concentration that originated from the Bellingshausen Sea sector (87.6 ± 110.0, 86.6 ±110.0, and 83.9 ± 109.0 ng m$^{-3}$ for 3-, 4-, and 5-day air mass back trajectories based estimates, respectively) was ~7 3-times higher than that which originated from the Weddell Sea sector (27.4 ± 19.3, 30.6 ± 27.5, and 33.9 ± 30.4 ng m$^{-3}$ for 3-, 4-, and 5-day air mass back trajectories based estimates, respectively) during the austral summer period in 2013–2014 (Fig. 3 4b). Although the period of satellite observations and in situ chemistry analysis is not exactly the same, both multi-year satellite-derived biological characteristics and aerosol chemistry data support the interpretation that there was higher abundance of DMS(P)-rich phytoplankton in the Bellingshausen Sea than in the Weddell Sea during the austral summer period (Fig. 2 3, Fig. 3 4, Figs. S3 S2, S4 S3 and S5 S4).

In the 8-year record, the monthly mean chlorophyll concentration was positively correlated with the monthly mean number concentration of nanoparticles for the air masses that originated from the Bellingshausen Sea in January and February ($r^2 = 0.69$, $n = 12$, $P < 0.05$; Fig. 4 5a). During this period, the contribution of the DMS(P)-rich phaeocystis to the chlorophyll concentration was highest in the Bellingshausen Sea (i.e., DMSP to chlorophyll ratio >100 mmol g$^{-1}$; dominance of phaeocystis >50%; dominance of diatoms <10%; Fig. 2 3, Fig. S4a S3a and Fig. S5 S4). Conversely, the increase in the chlorophyll concentration was not correlated with the increase in the number concentration of nanoparticles in the Weddell Sea (Fig. 4b 5b). As a consequence, the higher occurrence of nanoparticles from the Bellingshausen Sea inferred from our analysis was likely to be associated with a higher abundance of DMS(P)-rich phytoplankton, whereas the lower occurrence of nanoparticles from the Weddell Sea appeared to be associated with a higher abundance of DMS(P)-poor phytoplankton.

**4 Conclusions**

The physical properties of aerosol particles measured above the remote Antarctic Peninsula over 8 years were analyzed in conjunction with the satellite-derived biological characteristics around the observation site. These results show that the formation of nanoparticles was strongly associated not only with the biomass of phytoplankton but, more importantly, also its taxonomic composition in the Antarctic Ocean. Previous studies have reported that diatoms have a competitive advantage under conditions in which the mixed layers are shallow and the light levels are relatively high. Conversely, phaeocystis is well adapted to conditions in which mixed layers are deep and light levels are variable (e.g., Weber and El-Sayed, 1987; Arrigo et al., 1999; Goffart et al., 2000; Alvain et al., 2008; Arrigo et al., 2010). These results are consistent with the distribution of phytoplankton

**메모 포함[KTP26]:** (response to referee #2)
**Q3. Typical lifetime of PM is generally longer than 2 days:** We agree with referee #2 that the retention time of PM$_{10}$ is known to be >2 days in the lower troposphere. Therefore, we have applied 3-, 4- and 5-day air mass back trajectories to identify the potential origin of MSA.

**메모 포함[KTP27]:** (response to referee #2)
**Q3. Typical lifetime of PM is generally longer than 2 days:** We have estimated the MSA concentration originated from Bellingshausen Sea sector by applying 3-, 4- and 5-day air mass back trajectories.

**메모 포함[KTP28]:** (response to referee #2)
**Q3. Typical lifetime of PM is generally longer than 2 days:** We have estimated the MSA concentration originated from Weddell Sea sector by applying 3-, 4- and 5-day air mass back trajectories.

groups in the Bellingshausen and Weddell seas. Given that the mixed layer depth in the Bellingshausen Sea (45.6 ± 4.1 m) was relatively deeper than that of the Weddell Sea (36.2 ± 3.8 m; Fig. S6) during the austral summer period, the growth of DMS(P)-rich phaeocystis may therefore be more favorable in the Bellingshausen Sea. Sea-surface warming and freshening is commonly associated with a shallowing of the mixed layer depth (Capotondi et al., 2012). The warming trend has shown the spatial complexity across the Antarctic Ocean in recent decades (Turner et al., 2005). Therefore, all regions of the Antarctic Ocean will experience different changes in phytoplankton productivity and taxonomic composition in response to the climate change (Deppeler and Davidson, 2017). ~~Changes in the marine ecosystem could significantly influence the formation of aerosols and clouds in the remote marine atmosphere (Wang et al., 2018). Continuous measurements of the physiochemical properties of aerosols and key nucleation compounds are needed to improve our knowledge regarding the association between marine biota and the formation of aerosol particles and to understand their climate feedback roles in the remote marine environment.~~

In this study, we have focused on the relationship between the formation of nanoparticles and marine biota. The formation of secondary aerosols contributes significantly to the atmospheric aerosol number and accounts for half of the global cloud condensation nuclei (Merikanto et al., 2009; Sullivan et al., 2018). Our results indicate that changes in the taxonomic composition of marine phytoplankton (i.e., DMS(P)-rich species *vs.* DMS(P)-poor species) could have a significant impact on the aerosol properties in the remote marine environment. Precursors other than biogenic DMS could play a key role in the formation of new particles in the Antarctic atmosphere. In fact, 16 days of new particle formation events out of 101 events were observed when the air mass originated from the Antarctic Peninsula during the study period. Penguin colonies are dispersed throughout the Antarctic Peninsula (Croxall et al., 2002), and the emission of ammonia from these colonies could trigger the formation of nanoparticles (Weber et al., 1998; Croft et al., 2016). Moreover, iodine molecules produced by biotic and abiotic processes near sea-ice region are known to influence the formation of aerosol particles (Allan et al., 2015; Sipilä et al., 2016). Future studies are required to minimize the knowledge gaps related to multiple precursors and their source origins. Specifically, continuous measurements of the physiochemical properties of aerosol particles and molecular-scale measurements of chemical species (e.g., sulfur-, nitrogen-, and halogen-containing compounds) involved in nucleation processes are required to provide direct evidence for the contribution of these compounds to the formation and growth of aerosol particles and to understand their climate feedback roles in the remote marine environment.

**Author contributions**

KTP, JEH, JSK, TWK and YYJ designed the study. YYJ and YTG analyzed the physical properties of the aerosol particles. HSB, SB and RT operated the air sampler and analyzed the MSA. KTP and JEH wrote the manuscript.

**Acknowledgements**

This study was supported by the KOPRI project (PE18010, PE18140). We thank overwintering staff for assisting us in maintaining the aerosol equipment at the King Sejong Station.

메모 포함**[KTP29]:** (response to Short comment)
Q1. **Need to add explanation on the roles of iodine compounds in new particle formation events:** we have added a short paragraph indicating the roles of iodine-molecules on the formation of new particles and the limitation of our study.

메모 포함**[KTP30]:** (response to referee #2)
**Q1. Need to provide more comprehensive discussion on how this result build upon earlier findings and what exactly is the new scientific message:** In the revised manuscript, we have revised and added a paragraph explaining the key results of this study. Furthermore, the limitations of the present study and the scope of future studies have also been added.

(response to referee #3)
**Q1, Q3, and Q5. Present a whole analysis, including air masses originating other sectors:** we have a companion paper that describes the physical characteristics of the aerosol particles observed at the King Sejong station during the entire sampling period. The relationship between new particle formation and environmental parameters is complicated, owing to interplay among multiple sources and complicated processes. The number concentration of the nanoparticle was at its maximum during the productive summer period, and the frequency of new particle formation was highest when the air mass originated from the ocean domain. Therefore, we focused on the influence of marine biota on the formation of nanoparticles in this study. In the revised manuscript, we have added more description to clarify the key finding of this study. Furthermore, the limitations of the present study and the scope of future studies have also been added.

메모 포함[WU34]: (response to referee #1)
We have added this reference.

메모 포함[WU35]: (response to referee #1)
We have added this reference.

메모 포함[WU36]: (response to referee #2)
We have added this reference.

메모 포함[WU37]: (response to referee #2, 3 and Short comment)
We have added this reference.

메모 포함[WU38]: (response to referee #3)
We have added this reference.

메모 포함[WU39]: (response to referee #2, 3 and Short comment)
We have added this reference.

[revised manuscript text omitted]

메모 포함[WU42]: (response to referee #1) We have added this reference.

메모 포함[WU43]: (response to referee #2 and 3) We have added this reference.

메모 포함[WU44]: (response to referee #2) We have added this reference.

2013a.

[revised manuscript text omitted]

메모 포함[WU52]: (response to referee #2, 3 and Short comment) We have added this reference.

메모 포함[WU53]: (response to referee #1) We have added this reference.

[Figure]

**Figure 1:** (a) Hourly variations in the number concentration of particles >2.5 nm in diameter ($CN_{2.5}$, blue symbols) and the number concentration of particles >10 nm in diameter ($CN_{10}$, red symbols), (b) hourly variations in the number concentration of nanoparticles (ranging from 2.5 to 10 nm in diameter) calculated using the differences between $CN_{2.5}$ and $CN_{10}$, and (c) hourly variations in the retention time of 2-day air mass back trajectories over the three domains including ocean (blue), sea-ice (red), and land (black) domains from March 2009 to November 2016.

**메모 포함[WU54]:** (response to referee #3)
We have added new figure that represent the 8-year nanoparticle measurement and corresponding air mass transport history.

[Figure]

**Figure  2:** (a) Back-trajectories of the air masses arriving at King Sejong Station (62.2°S, 58.8°W; star symbol), Antarctic Peninsula. The colors indicate the percentage (%) of the air mass located at that spot during the 2 days prior to arriving at the observation site. Note that the air mass back-trajectories that did not have >90% retention in the two selected ocean domains (i.e., Bellingshausen and Weddell Seas) were excluded. (b) Seasonal variation of nanoparticles (2.5–10 nm in diameter, $CN_{2.5-10}$) observed at King Sejong Station between March 2009 and December 2016. Blue and red symbols indicate the number concentration of nanoparticles that originated from the Bellingshausen and Weddell seas, respectively. The error bars indicate the 95% confidence interval estimated by bootstrap method from the monthly $CN_{2.5-10}$ data.

[Figure]

Figure 2 3: (a) Monthly mean chlorophyll concentration during the months of December, January, and February in 2009–2016, (b) monthly mean DMSP concentration during the months of December, January, and February in 2009–2016, (b c) phytoplankton taxonomic composition including diatoms (DIA), Prochlorococcus (PRO), nanoeucaryotes (NEU), Synechococcus (SLC) and phaeocystis (PHA) during the months of December, January and February in 1997 2010 estimated using the PHYSAT method with the climatology map obtained from SeaWiFS archive, and (e d) monthly mean DMSP to chlorophyll ratio during the months of December, January, and February in 2009–2016. Note that the phytoplankton taxonomic composition was estimated using the SeaWiFS (Sea-viewing Wide Field-of-View Sensor) ocean color sensor which is available from September 1997 to December 2010.

메모 포함[KTP55]: (response to referee #2)
Q2. **Need to compare the absolute DMSP concentration:** As this referee suggested, we have added figure for sea-surface DMSP concentration.

메모 포함[KTP56]: (response to referee #1)
**Q1. How representative is the PHYSAT database for the period 1997-2010 with regard to your observation period?** We have replaced "*during the months of December, January and February in 1997–2010*" to "*estimated using the PHYSAT method with the climatology map obtained from SeaWiFS archive*"to clarify the meaning.

메모 포함[KTP57]: (response to referee #1)
**Q1. How representative is the PHYSAT database for the period 1997-2010 with regard to your observation period?** We have removed this sentence and more detailed description for the SeaWiFS archive were added in chapter 2.3 and 3.2.

[Figure]

**Figure 4:** (a) Daily concentration of MSA collected at the sampling site during the summer periods in 2013 and 2014 (explicitly, from 14 January to 28 February in 2013, and from 2 December 2013 to 18 January 2014). (b) The mean MSA concentration that potentially originated from the Bellingshausen Sea sector (<58.8°W) and the Weddell Sea sector (>58.8°W) estimated by applying 3-, 4-, and 5-day air mass back trajectories during the austral summer periods in 2013 and 2014. The error bars indicate 1 standard deviation (1σ) from the mean values.

메모 포함[KTP58]: (response to referee #1)
**Q5. Provide daily MSA concentration data:** We have added daily MSA concentration data in the revised manuscript.

메모 포함[KTP59]: (response to referee #2)
**Q2. Typical lifetime of PM is generally longer than 2 days:** We have estimated the MSA concentration originated from the Bellingshausen Sea and Weddell Sea sectors by applying 3, 4 and 5 days air mass back trajectories.

[Figure]

**Figure 4 5:** (a) Relationship between the monthly mean chlorophyll concentration for the Bellingshausen Sea (55°S–65°S, 60°W–80°W) and the monthly mean number concentration of nanoparticles that originated from the Bellingshausen Sea in 2009–2016. (b) Relationship between the monthly mean chlorophyll concentration for the Weddell Sea (55°S–65°S, 40°W–60°W) and the monthly mean number concentration of nanoparticles that originated from the Weddell Sea in 2009–2016. The filled blue, filled red and open red symbols indicate the data obtained in December, January and February, respectively. The solid lines represent the best fit.

*Supplement for*

**New particle formation events observed at the King Sejong Station, Antarctic Peninsula – Part 2: Link with the oceanic biological activities**

Eunho Jang et al.

*Correspondence to*: Ki-Tae Park (ktpark@kopri.re.kr)

**Calculation of the sea surface DMSP concentration**

 Several studies reported that DMSP and DMS were strongly linked to with several environmental parameters such as solar radiation, sea-surface temperature, and mixing state of the sea-surface (Vallina and Simo, 2007). *Gali et al.* (2015) developed a DMSP algorithm based on satellite-derived chlorophyll (to measure phytoplankton biomass) and the light exposure regime (to measure key environmental factors controlling DMSP production; i.e., solar radiation and mixing state). In this algorithm, euphotic layer depth ($Z_{eu}$) and mixed layer depth (MLD) dataset were applied to establish a mixing state of the sea surface (stratified vs. mixed water column), and the variability in modeled and measured DMSP was improved by adding sea-surface temperature and $\log_{10}(Zeu/MLD)$ as predictors for the stratified and mixed subsets in the proposed algorithm. Additionally, a sub-model based on particulate inorganic carbon (PIC) was developed to complement DMSPt diagnosis in coccolithophore blooms, where satellite chlorophyll concentration may not be reliable. The database was divided into three subsets including 'stratified water ($Z_{eu}$/MLD > 1)', 'mixed water ($Z_{eu}$/MLD < 1)' and 'undefined water ($Z_{eu}$ or MLD is unavailable)' based on the ratio between the euphotic layer depth ($Z_{eu}$) and the mixed layer depth (MLD). The $DMSP_t$ concentrations in stratified, mixed and undefined water were calculated using Equations (S1), (S2) and (S3), respectively:

$$\mathrm{Log}_{10}DMSP_t = 1.70 + 1.14\log_{10}Chl_t + 0.44\log_{10}Chl_t^2 + 0.063SST - 0.0024SST^2 \tag{S1}$$

$$\mathrm{Log}_{10}DMSP_t = 1.74 + 0.81\log_{10}Chl_t + 0.60\log_{10}(Z_{eu}/MLD) \tag{S2}$$

$$\log_{10}DMSP_t = -1.052 - 3.185\log_{10}PIC - 0.783(\log_{10}PIC)^2 \tag{S3}$$

The level-3 product of the Moderate Resolution Imaging Spectroradiometer on the Aqua (MODIS-Aqua) satellites was used for the chlorophyll concentration ($Chl_t$), sea surface temperature at nighttime (SST) and the calcite concentration (PIC). The monthly mixed layer depth (MLD) was retrieved by Monthly Isopycnal and Mixed-layer Ocean Climatology (MIMOC) at a resolution of 0.5°. All of the MODIS-Aqua products at a resolution of 4 km were averaged onto a 0.5° interval grid of MIMOC climatology to run the DMSPt algorithm. The euphotic layer depth ($Z_{eu}$) was calculated using satellite-derived chlorophyll data as shown in Equation (S4) (Morel et al., 2007).

$$\log_{10}Z_{eu} = 1.524 - 0.436\log_{10}Chl_t - 0.0145(\log_{10}Chl_t)^2 + 0.0186(\log_{10}Chl_t)^3 \tag{S4}$$

메모 포함[KTP60]: (response to referee #1)
**Q1. What is the impact of SST, MLD and PIC compared to Chl?**
We have added detailed explanation regarding the DMSP algorithm in the revised supplementary.

**Table S1.** Monthly average, 1 standard deviation and 95% confidence interval of nanoparticles (2.5–10 nm in diameter, $CN_{2.5-10}$) that originated from the Bellingshausen and Weddell Seas during the study period. A $t$-test was used to determine if there is a statistically significant difference between the means number concentration nanoparticles originated from the two selected ocean domains.

| | | Avg. | Std. | 95% confidence interval* | | $p$-value ($t$-test) |
| --- | --- | --- | --- | --- | --- | --- |
| | | | | Upper bound | Lower bound | |
| Jan. | Weddell sea | 332.0 | 920.7 | 196.3 | 782.4 | 0.0003 |
| | Bellingshausen sea | 835.9 | 2673.2 | 705.9 | 1004 | |
| Feb. | Weddell sea | 254.3 | 284.0 | 181.4 | 355.8 | 0.0010 |
| | Bellingshausen sea | 523.7 | 2130.7 | 417.2 | 695.4 | |
| Mar. | Weddell sea | 60.7 | 60.3 | 47.0 | 78.2 | < 0.0001 |
| | Bellingshausen sea | 166.6 | 550.3 | 142.6 | 208.5 | |
| Apr. | Weddell sea | 70.0 | 103.6 | 53.9 | 96.4 | 0.0245 |
| | Bellingshausen sea | 100.9 | 272.4 | 87.7 | 126.6 | |
| May | Weddell sea | 89.6 | 74.5 | 75.2 | 108.4 | < 0.0001 |
| | Bellingshausen sea | 45.2 | 56.2 | 41.0 | 50.9 | |
| Jun. | Weddell sea | 58.0 | 22.0 | NaN** | NaN | NaN |
| | Bellingshausen sea | 57.7 | 62.1 | 52.7 | 64.1 | |
| Jul. | Weddell sea | 22.9 | 17.9 | 15.1 | 35.1 | 0.0031 |
| | Bellingshausen sea | 42.8 | 56.8 | 36.8 | 51.6 | |
| Aug. | Weddell sea | - | - | NaN | NaN | NaN |
| | Bellingshausen sea | 58.3 | 78.6 | 47.5 | 73.6 | |
| Sep. | Weddell sea | 3.7 | - | NaN | NaN | NaN |
| | Bellingshausen sea | 97.9 | 85.9 | 91.0 | 105.5 | |
| Oct. | Weddell sea | 193.1 | 160.5 | NaN | NaN | NaN |
| | Bellingshausen sea | 129.0 | 405.1 | 110.3 | 197.7 | |
| Nov. | Weddell sea | 88.0 | 61.0 | 74.3 | 107.4 | < 0.0001 |
| | Bellingshausen sea | 176.7 | 331.9 | 154.0 | 214.3 | |
| Dec. | Weddell sea | 200.5 | 380.5 | 56.5 | 499.9 | 0.2111 |
| | Bellingshausen sea | 343.0 | 1138.8 | 277.6 | 449.0 | |

*confidence interval was estimated by bootstrap method that was calculated from 10,000 subsamples generated by random sampling with replacement from monthly $CN_{2.5-10}$ data.
**number of monthly $CN_{2.5-10}$ data <10 was excluded from the bootstrap and $t$-test.

[Figure]

**Figure S1:**

메모 포함**[WU61]:** (response to referee #3)
We have moved Figure S1 to Figure 1a in the main manuscript.

[Figure]

(a)      (b)

Sea ice    Land    Ocean

**Figure**  **S1:** Percentage of the hourly trajectory points that passed over the three major areas surrounding the observation site including sea-ice (red), land (yellow) and ocean (blue) to the total number of hourly trajectory points in the 2-day air-mass trajectory during (a) the overall period (from January to December) and (b) the austral summer period (December, January and February) between March 2009 and November 2016.

[Figure]

**Figure S3 S2:** (a) Monthly mean chlorophyll concentration around the observation site between 2009 and 2016 (55ºS–65ºS, 40ºW–80ºW). (b) Monthly mean chlorophyll concentration for the two selected ocean domains including the Weddell (red symbols; 55ºS–65ºS, 40ºW–60ºW) and Bellingshausen (blue symbols; 55ºS–65ºS, 60ºW–80ºW) seas during the phytoplankton bloom period (October–February). Note that the monthly mean chlorophyll concentration was not available from May to August due to insufficient satellite-derived values (less than 10%) during the austral winter period.

[Figure]

**Figure**  S3: The percentage of the dominant phytoplankton groups in the two ocean domains including (a) the Bellingshausen and (b) Weddell seas estimated using the PHYSAT method with the climatology map obtained from SeaWiFS archive during the austral summer period

메모 포함**[KTP62]:** (response to referee #1)
**Q1. How representative is the PHYSAT database for the period 1997-2010 with regard to your observation period?** We have replaced "*using the SeaWiFS archive (from September 1997 to December 2010) between 1997 and 2010*" to "*using the PHYSAT method with the climatology map obtained from SeaWiFS archive during the austral summer period*"to clarify the meaning.

[Figure]

**Figure**  **S4:** (a) Monthly mean DMSP concentration and (b)  monthly mean DMSP to chlorophyll ratio in the Bellingshausen (blue bars) and Weddell (red bars)  Seas during the austral summer period between March 2009 and November 2016.

메모 포함**[WU63]:** (response to referee #2)
Q2. **Need to compare the absolute DMSP concentration:** As this referee suggested, we have added figure for sea surface DMSP concentration.

[Figure]

**Figure  S5:** Mixed layer depth retrieved using the Monthly Isopycnal and Mixed-layer Ocean Climatology (MIMOC) during the austral summer period surrounding King Sejong Station (red star symbol).

---

## Author Response (AR2)

**Response to Referee 2**

We thank Referee 2 for providing insightful comments. Our changes in response to this Referee's suggestions are provided below.

**Minor comments**

1. **The third sentence of section 3.1 (The physical…) is a bit unclear:** We have modified the third sentence of section 3.1 (*The physical aspect of…*) to "*A detailed explanation of physical characteristics of new particle formation events (e.g., frequency, formation rate, and growth rate) at King Sejong Station during the same period is explained in Kim et al. (2018).*" (P5, lines 27–29) in the revised manuscript to clarify the meaning.

2. **The authors report the particle growth rates (section 2.2) and concentrations (section 3.1) with a too high accuracy. I doubt that the particle growth rate can be determined with an accuracy better than 0.1nm/h, or that the particle could be known with an accuracy of 5 digits:** As this referee suggested, we have changed "*0.68 ± 0.27*" to "*approximately 0.7 ± 0.3*" (P4, line 10), and marked the number concentration of particles with an integer (P5, lines 15–19) in the revised manuscript.

3. **Section 3.2, last paragraph: it seems that "by" is missing from "…the DMSP cleavage enzyme":** We have added "*by*" in the revised manuscript (P7, line 18).

4. **In the last paragraph of section 4, the authors state that secondary aerosol contributed half of the global CCN. This far too strictly said. There are a number global modeling papers on this subject they all give a relative broad range of potential values for the percentage of CCN due to atmospheric new particle formation and growth:** We have changed "*half'*" to "*30–80%*" in the revised manuscript. Furthermore, we have added more references (Westervelt et al., 2014, ACP; Sanchez et al., 2018, Sci. Rep.) that report the contribution of secondary aerosols to CCN.

[revised manuscript text omitted]

메모 포함[KTP3]: Response to referee #2
Q2. We have changed "835.9 ± 2673.2" to "836 ± 2673".

메모 포함[KTP4]: Response to referee #2
Q2. We have changed "332.0 ± 920.7" to "332 ± 921".

메모 포함[KTP5]: Response to referee #2
Q2. We have changed "567.5 ± 249.3" to "568 ± 249".

메모 포함[KTP6]: Response to referee #2
Q2. We have changed "262.3 ± 66.1" to "262 ± 66".

메모 포함[KTP7]: Response to referee #2
Q2. We have changed "97.2 ± 51.1" to "97 ± 51".

메모 포함[KTP8]: Response to referee #2
Q2. We have changed "73.2 ± 56.9" to "73 ± 57".

[revised manuscript text omitted]

메모 포함[WU10]: (response to referee #2)
**Q4.** we have changed "*half*" to "*30–80%*". Furthermore, we have added more references that report the contribution of secondary aerosols to CCN.

[revised manuscript text omitted]

Several studies reported that DMSP and DMS were strongly linked to several environmental parameters such as solar radiation, sea-surface temperature, and mixing state of the sea surface (Vallina and Simo, 2007). *Gali* et al. (2015) developed a DMSP algorithm based on satellite-derived chlorophyll (to measure phytoplankton biomass)

5 and the light exposure regime (to measure key environmental factors controlling DMSP production). In this algorithm, euphotic layer depth ($Z_{eu}$) and mixed layer depth (MLD) dataset were applied to establish a mixing state of the sea surface (stratified vs. mixed water column), and the variability in modeled and measured DMSP was improved by adding sea-surface temperature and $\log_{10}(Zeu/MLD)$ as predictors for the stratified and mixed subsets in the proposed algorithm. Additionally, a sub-model based on particulate inorganic carbon (PIC) was

10 developed to complement DMSP diagnosis in coccolithophore blooms, where satellite chlorophyll concentration may not be reliable. The database was divided into three subsets including 'stratified water ($Z_{eu}$/MLD > 1)', 'mixed water ($Z_{eu}$/MLD < 1)' and 'undefined water ($Z_{eu}$ or MLD is unavailable)' based on the ratio between the euphotic layer depth ($Z_{eu}$) and the mixed layer depth (MLD). The $DMSP_t$ concentrations in stratified, mixed and undefined water were calculated using Equations (S1), (S2) and (S3), respectively:

15 $\text{Log}_{10}\text{DMSP}_t = 1.70 + 1.14\log_{10}\text{Chl}_t + 0.44\log_{10}\text{Chl}_t^2 + 0.063\text{SST} - 0.0024\text{SST}^2$  (S1)
$\text{Log}_{10}\text{DMSP}_t = 1.74 + 0.81\log_{10}\text{Chl}_t + 0.60\log_{10}(Z_{eu}/\text{MLD})$  (S2)
$\log_{10}\text{DMSP}_t = -1.052 - 3.185\log_{10}\text{PIC} - 0.783(\log_{10}\text{PIC})^2$  (S3)

tThe level-3 product of the Moderate Resolution Imaging Spectroradiometer on the Aqua (MODIS-Aqua) satellites was used for the chlorophyll concentration ($\text{Chl}_t$), sea surface temperature at nighttime (SST) and the

20 calcite concentration (PIC). The monthly mixed layer depth (MLD) was retrieved by Monthly Isopycnal and Mixed-layer Ocean Climatology (MIMOC) at a resolution of 0.5°. All of the MODIS-Aqua products at a resolution of 4 km were averaged onto a 0.5° interval grid of MIMOC climatology to run the DMSPt algorithm. The euphotic layer depth ($Z_{eu}$) was calculated using satellite-derived chlorophyll data as shown in Equation (S4) (Morel et al., 2007).

25 $\log_{10}Z_{eu} = 1.524 - 0.436\log_{10}\text{Chl}_t - 0.0145(\log_{10}\text{Chl}_t)^2 + 0.0186(\log_{10}\text{Chl}_t)^3$  (S4)

**Table S1.** Monthly average, 1 standard deviation and 95% confidence interval of nanoparticles (2.5–10 nm in diameter, $CN_{2.5-10}$) that originated from the Bellingshausen and Weddell Seas during the study period. A $t$-test was used to determine if there is a statistically significant difference between the means number concentration nanoparticles originated from the two selected ocean domains.

| | | Avg. | Std. | 95% confidence interval* | | $p$-value ($t$-test) |
| | | | | Upper bound | Lower bound | |
|---|---|---|---|---|---|---|
| Jan. | Weddell sea | 332.0 | 920.7 | 196.3 | 782.4 | 0.0003 |
| | Bellingshausen sea | 835.9 | 2673.2 | 705.9 | 1004 | |
| Feb. | Weddell sea | 254.3 | 284.0 | 181.4 | 355.8 | 0.0010 |
| | Bellingshausen sea | 523.7 | 2130.7 | 417.2 | 695.4 | |
| Mar. | Weddell sea | 60.7 | 60.3 | 47.0 | 78.2 | < 0.0001 |
| | Bellingshausen sea | 166.6 | 550.3 | 142.6 | 208.5 | |
| Apr. | Weddell sea | 70.0 | 103.6 | 53.9 | 96.4 | 0.0245 |
| | Bellingshausen sea | 100.9 | 272.4 | 87.7 | 126.6 | |
| May | Weddell sea | 89.6 | 74.5 | 75.2 | 108.4 | < 0.0001 |
| | Bellingshausen sea | 45.2 | 56.2 | 41.0 | 50.9 | |
| Jun. | Weddell sea | 58.0 | 22.0 | NaN** | NaN | NaN |
| | Bellingshausen sea | 57.7 | 62.1 | 52.7 | 64.1 | |
| Jul. | Weddell sea | 22.9 | 17.9 | 15.1 | 35.1 | 0.0031 |
| | Bellingshausen sea | 42.8 | 56.8 | 36.8 | 51.6 | |
| Aug. | Weddell sea | - | - | NaN | NaN | NaN |
| | Bellingshausen sea | 58.3 | 78.6 | 47.5 | 73.6 | |
| Sep. | Weddell sea | 3.7 | - | NaN | NaN | NaN |
| | Bellingshausen sea | 97.9 | 85.9 | 91.0 | 105.5 | |
| Oct. | Weddell sea | 193.1 | 160.5 | NaN | NaN | NaN |
| | Bellingshausen sea | 129.0 | 405.1 | 110.3 | 197.7 | |
| Nov. | Weddell sea | 88.0 | 61.0 | 74.3 | 107.4 | < 0.0001 |
| | Bellingshausen sea | 176.7 | 331.9 | 154.0 | 214.3 | |
| Dec. | Weddell sea | 200.5 | 380.5 | 56.5 | 499.9 | 0.2111 |
| | Bellingshausen sea | 343.0 | 1138.8 | 277.6 | 449.0 | |

*confidence interval was estimated by bootstrap method that was calculated from 10,000 subsamples generated by random sampling with replacement from monthly $CN_{2.5-10}$ data.
**number of monthly $CN_{2.5-10}$ data <10 was excluded from the bootstrap and $t$-test.

[Figure]

**Figure S1:** Percentage of the hourly trajectory points that passed over the three major areas surrounding the observation site including sea-ice (red), land (yellow) and ocean (blue) to the total number of hourly trajectory points in the 2-day air-mass trajectory during (a) the overall period (from January to December) and (b) the austral summer period (December, January and February) between March 2009 and November 2016.

[Figure]

**Figure S2:** (a) Monthly mean chlorophyll concentration around the observation site between 2009 and 2016 (55ºS–65ºS, 40ºW–80ºW). (b) Monthly mean chlorophyll concentration for the two selected ocean domains including the Weddell (red symbols; 55ºS–65ºS, 40ºW–60ºW) and Bellingshausen (blue symbols; 55ºS–65ºS, 60ºW–80ºW) seas during the phytoplankton bloom period (October–February). Note that the monthly mean chlorophyll concentration was not available from May to August due to insufficient satellite-derived values (less than 10%) during the austral winter period.

[Figure]

**Figure S3:** The percentage of the dominant phytoplankton groups in the two ocean domains including (a) the Bellingshausen and (b) Weddell seas estimated using the PHYSAT method with the climatology map obtained from SeaWiFS archive during the austral summer period.

[Figure]

**Figure S4:** (a) Monthly mean DMSP concentration and (b) monthly mean DMSP-to-chlorophyll ratio in the Bellingshausen (blue bars) and Weddell (red bars) Seas during the austral summer period between March 2009 and November 2016.

[Figure]

**Figure S5:** Mixed layer depth retrieved using the Monthly Isopycnal and Mixed-layer Ocean Climatology
(MIMOC) during the austral summer period surrounding King Sejong Station (red star symbol).